# Pathogen-specific social immunity is associated with erosion of individual immune function in an ant

Florent Masson [1] ✉, Rachael Louise Brown[1], Joel Vizueta [2], Thea Irvine[1], Zijun Xiong [3], Jonathan Romiguier [4] & Nathalie Stroeymeyt [1] ✉

Contagious diseases are a major threat to societies in which individuals live in close contact. Social insects have evolved collective defense behaviors, such as social care or isolation of infected workers, that prevent outbreaks of pathogens. It has thus been suggested that individual immunity is reduced in species with such 'social immunity'. However, this hypothesis has not been tested functionally. Here, we characterize the immune response of the ant *Lasius niger* using a combination of genomic analysis, experimental infections, gene expression quantification, behavioural observations and pathogen quantifications. We uncover a striking specialization of immune responses towards different pathogens. Systemic individual immunity is effective against opportunistic bacterial infections, which are not covered by social immunity, but is not elicited upon fungal infections, which are effectively controlled by social immunity. This specialization suggests that immune layers have evolved complementary functions predicted to ensure the most cost-effective response against a wide range of pathogens.

A crucial challenge for complex societies is to limit the transmission of contagious diseases despite extensive contacts among individuals. In most Metazoan species, this is achieved via an immune system that fights infections and promotes resilience at the individual level ('individual immunity'). In addition, some highly social species have evolved both prophylactic and pathogen-induced collective traits that reduce the risk of pathogen transmission between individuals[1,2] and confer a society-level protection termed 'social immunity'[1,3]. Social immunity has been reported in numerous species including humans, bats, guppies, house finches and social insects[2], but the trade-offs and interplays between individual and social defenses remain poorly understood[4].

Social immunity mechanisms have been best characterized in social insects (ants, termites and some bees and wasps)[1], which use an array of social defenses, such as the rearrangement of social interaction networks[5,6], isolation of contaminated workers[7], or destruction of infected brood[8]. Because of the remarkable effectiveness of social

immunity mechanisms at preventing disease outbreaks, it has been suggested that highly social species may no longer require strong individual immunity[1,4]. However, there is conflicting evidence for this hypothesis. Some genomic studies reported a reduced complexity of the main systemic immune pathways of insects (NF-κB pathways Toll and Immune Deficiency IMD) in Apidae and Termitoidea[9,10], but these findings were not complemented with functional investigations. Furthermore, the reported simplification mostly affects regulatory and effector genes, while the core components of these pathways are conserved and show signs of positive selection across social insects[9,10]. As the expression of some immune genes is induced in ant queens and workers upon immune challenges[11,12], this suggests that individual immunity could be functional in these species. Here we hypothesize that individual immunity in social insect species have evolved towards a reduced, but still functional arsenal specialized in fighting microbes that are not targeted by social immunity.

[1]School of Biological Sciences, University of Bristol, Bristol, UK. [2]Villum Centre for Biodiversity Genomics, Section for Ecology and Evolution, Department of Biology, University of Copenhagen, Universitetsparken 15, Copenhagen, Denmark. [3]BGI Research, Wuhan, China. [4]ISEM, University of Montpellier, CNRS, IRD, Montpellier, France. ✉e-mail: florent.masson@bristol.ac.uk; nathalie.stroeymeyt@bristol.ac.uk

To test this hypothesis, we used genomic analysis, experimental infections, and behavioral assays to examine the level of protection conferred by the two main insect systemic immune pathways (NF-κB pathways Toll and IMD) in a model social insect species known to benefit from strong social immunity against fungal infections, the black garden ant *Lasius niger*. Our results indicate that *L. niger* NF-κB pathways are functional and protect individuals against opportunistic bacterial infections that fall outside the scope of social immunity, but play little role against fungal infections. This supports the hypothesis that the two lines of defense (individual and social) are not redundant but complementary and provide targeted protection against a wide spectrum of pathogens.

## Results

### The genome of *L. niger* encodes complete NF-κB pathways

We first assessed the genomic erosion of *L. niger* NF-κB pathways IMD and Toll. In the reference model *Drosophila melanogaster*, the IMD pathway mainly responds to peptidoglycan (PGN) from Gram-negative bacteria, while the Toll pathway mainly responds to PGN from Gram-positive and some Gram-negative bacteria, and β-glucans from fungal cell walls[13,14] (Fig. 1). Although variability exists, the core components of these pathways are largely conserved across insects, including Hymenoptera such as bumble bees, honeybees and ants[10,15,16].

The *L. niger* genome encodes homologues of all key components of the IMD pathway (Fig. 1 and Supplementary Table 1) except IKKγ. Its presence was deemed uncertain because *D. melanogaster*'s IKKγ does not have a convincing homologue in the *L. niger* genome, but we identified a candidate protein with 'NF-κB essential modulator' functional domains that could fulfill IKKγ's function in *L. niger* (Supplementary Table 1), suggesting that the IMD pathway is structurally complete in this species. The *L. niger* genome also encodes homologues of all genes participating in the intracellular part of the Toll pathway, involved in both immunity and embryonic development (Fig. 1 and Supplementary Table 2), except for *dif*, which is absent in most insect taxa[15]. In *Drosophila*, the extracellular part of the pathway is immunity-specific and recognizes microbes through the pattern-recognition receptors GNBP3 for fungal β-glucans and GNBP1 and PGRP-SA for bacterial (lys)-type PGN[14]. The genome of *L. niger* codes for one complete PGRP-SA gene, one complete GNBP gene and one pseudogenized GNBP copy with a premature stop codon, possibly arisen from a duplication (Supplementary Fig. 1A). Both *L. niger* GNBPs are orthologs of the honeybee *Apis mellifera* GNBP1-2, which belongs to an orthogroup conserved across Hymenoptera (Supplementary Fig. 1B). Within this orthogroup, *Lasius flavus* and *Lasius neglectus* also have one complete and one partial copy whereas other species such as *Temnothorax* sp., *Solenopsis* sp. and *Monomorium* sp. have two complete copies. We found a second group of GNBPs in Hymenoptera, represented by GNBP1-1 in *A. mellifera*. This second orthogroup is detected in Formicinae such as *Camponotus floridanus*, but not in *Lasius niger* (Supplementary Fig. 1B). This indicates that *Lasius niger* has a reduced number of functional GNBP-coding genes compared to other ant species.

Collectively, these data suggest that both pathways in *L. niger* are structurally similar to that of other Hymenoptera[10,15,16]. However, while the IMD pathway appears complete and thus may be fully functional, the reduced number of GNBP coding genes compared to other Hymenoptera and the pseudogenization of one copy raise the question of whether the *L. niger* Toll pathway can properly detect fungi.

### *L. niger* individual immunity relies on a reduced set of antimicrobial effectors

We next searched for genes encoding antimicrobial peptides (AMPs), the main immune effectors under the control of the NF-κB pathways[17]. AMPs are short cationic peptides secreted in large quantities by the fat body to target invading microbes. The selective activation of Toll or IMD pathways leads to the activation of a subset of AMP genes adapted to the class of pathogen encountered[14]. In the *L. niger* genome, we detected only four genes coding for AMPs presumably involved in systemic immunity (Supplementary Table 3): one Hymenoptaecin (Hym), two Defensins (Def1 and 2) and one Crustin (Cru). This immune arsenal is at the low end of that described in other ant species[18], and far below the 20 and 44 AMP repertoires described in *D. melanogaster* and the non-social hymenopteran *Nasiona vitripennis*, respectively[15].

Hymenoptaecins are broad-spectrum AMPs specific to Hymenoptera. In *Apis mellifera*, Hym is active against various Gram-positive and Gram-negative bacteria[19]. In *L. niger*, the *hym* promoter contains up to 4 binding sites for Relish (the IMD pathway transcription factor) and 2 binding sites for Dorsal (the main Toll pathway transcription factor), suggesting a control by the IMD pathway with a possible influence of the Toll pathway (Supplementary Fig. 2A).

Defensins are widespread across metazoans and primarily active against Gram-positive bacteria[20], with occasional activity against Gram-negative bacteria and fungi[17,21]. Viral infections can elicit *Drosophila def*, but the peptide does not have antiviral activity[22]. Furthermore, some Defensins promote wound healing independently of infection, both in mammals and in insects[23,24]. The *L. niger def1* promoter does not have any predicted NF-κB binding site, which suggests that its regulation is independent of these pathways. The *L. niger def2* promoter carries two predicted Dorsal binding sites, suggesting Toll-dependent regulation (Supplementary Fig. 2B). Despite insect Defensins having strongest activity against Gram-positive bacteria, their gene expression is usually controlled by the IMD pathway[17] and, accordingly, the promoters of *D. melanogaster* and *A. mellifera* carry predicted Relish binding sites. This contrasts with the *L. niger def1 and def2* promoters which do not carry any (Supplementary Fig. 2B).

Crustins are a large family of AMPs primarily found in Crustaceans[25]. The recently discovered hymenopteran Crustins have not been functionally characterized and their biological function remains hypothetical. In *L. niger*, the *cru* promoter does not carry any predicted Relish-binding site and only one Dorsal-binding site, suggesting weak regulation by the Toll pathway or NF-κB-independent regulation (Supplementary Fig. 2C).

Finally, we detected a sequence coding for a Waprin, an antimicrobial peptide closely related to Crustins. As Waprins are typically identified as a venom component rather than as effectors of systemic immunity, it was not considered further in this study[26,27].

In summary, our analysis identified a single AMP coding gene, *hym*, predicted to be under strong NF-κB transcriptional control, and three genes, *def1*, *def2* and *crustin*, with a weaker or non-existent NF-κB transcriptional control.

### A single antimicrobial peptide is elicited by bacterial but not fungal infections

To experimentally assess the ability of these AMPs to respond to microbial infections, we injected different microbes into the gaster (abdomen) of *L. niger* workers: *Erwinia carotovora carotovora 15* (*Ecc15*), a Gram-negative bacterium expected to elicit the IMD pathway[28], *Micrococcus luteus*, a Gram-positive bacterium expected to elicit the Toll pathway[29,30], and conidiospores of *Metarhizium brunneum*, a generalist entomopathogenic fungus expected to elicit the Toll pathway[31].

Comparison of gene expression between unchallenged and sham-treated ants (injected with sterile buffer) revealed a significant AMP induction by tissue damage (sometimes referred to as 'sterile immunity'[32]) for *hym* and *def1*, but not *def2* or *cru* (Fig. 2A; Linear Mixed-effects Model (LMM); post-hoc comparisons with Benjamini-Hochberg correction for multiple comparisons, *hym*: z = −23.59, p < 0.0001; *def1*: z = −18.48, p < 0.0001; *def2*: z = 1.84, p = 0.066; *cru*: z = −0.54, p = 0.593). *Hym* was further induced by both bacteria compared to sham treatment (z < −5.28, p < 0.0001 for both comparisons),

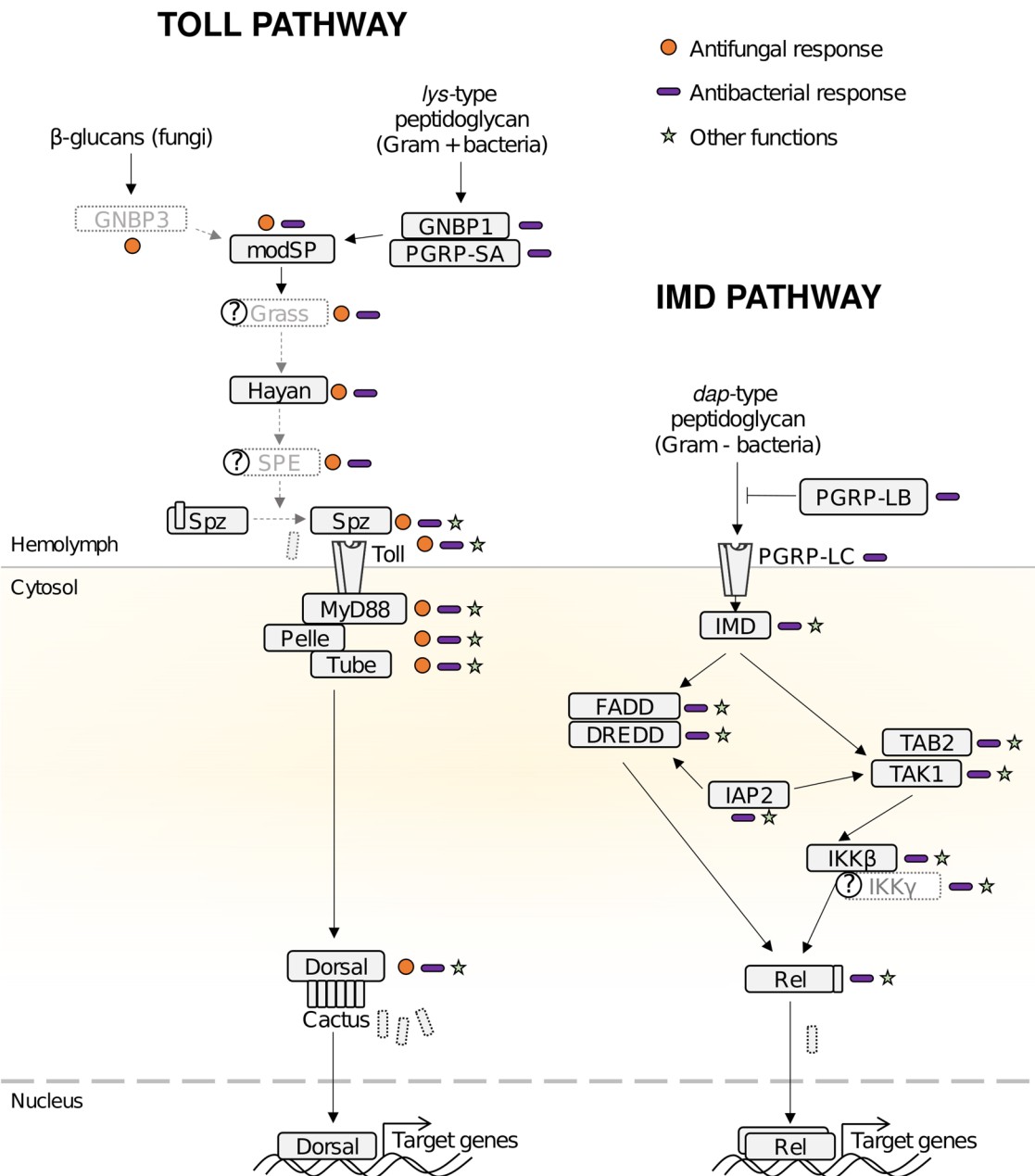

**Fig. 1 | *L. niger* NF-κB immune pathways.** *D. melanogaster* NF-κB immune pathways were used as reference. The IMD pathway is elicited by DAP-type peptidoglycan (PGN), a cell wall component of all Gram-negative and some Gram-positive bacteria. In Drosophila, IMD activation is mediated by PGN recognition by the membrane receptor PGRP-LC, which triggers an intracellular signaling cascade by activating the death-domain containing protein Imd. Imd complexes with FADD and the caspase DREDD and is activated by cleavage and ubiquitination mediated by DREDD and Iap2, respectively. This leads to the recruitment of a Tab2/Tak1 complex, which recruits an IKK complex, which in turn leads to the activating cleavage of the NF-κB transcription factor Relish. Cleaved Relish dimerizes, translocates into the nucleus, and regulates target gene expression. The Toll pathway is elicited by lysine (lys)-type PGN found in the cell wall of most Gram-positive bacteria, β-glucans from fungal cell walls, and fungus-specific protease activities (not shown on the figure). The binding of lys-type PGN and β-glucans to their cognate receptor triggers an extracellular enzymatic cascade that builds on the activating cleavage of serine proteases including ModSP, Grass, Hayan, and Spätzle-Processing-Enzyme (SPE). The cascade culminates with the cleavage of Spätzle (Spz) by SPE and its binding to the membrane receptor Toll. The intracellular signaling cascade downstream of Toll triggers the degradation of the NF-κB inhibitor Cactus, which releases the transcription factors Dorsal and Dorsal-Interacting Factor (DIF), which translocate into the nucleus and regulate target gene expression. Boxes with black plain lines indicate the presence of a homologue in the *L. niger* genome. Grey boxes with dotted lines indicate the absence of homologue. Question marks indicate uncertainties (presence of genes from the same family and with significant homology, but no clear identification). Arrows indicate protein-protein interactions as reported in the *D. melanogaster* literature. Circles indicate a gene involved in antifungal immunity, rods in antibacterial immunity, and stars denote non-immune functions. See Supplementary Table 1–3 for a complete gene list.

with a non-significant trend towards a stronger response to Gram-negative bacteria (*Ecc15* vs *M. luteus*: $z = -1.94$; $p = 0.062$). *Def2* was not induced by any challenge. Instead, it showed significant downregulation after injection of *Ecc15* ($z = 20.55$, $p < 0.0001$) and a non-significant downward trend after injection of *M. luteus* ($z = 1.89$, $p = 0.066$). *Def1* and *cru* expression did not differ between sham-treated and bacteria-challenged individuals ($|z| < 2.32$, $p > 0.102$ for all comparisons). Overall, these results indicate that *hym* responds to both tissue damage and bacterial cues, while *def1* responds to tissue damage only. *Def2* and *cru* are respectively downregulated or

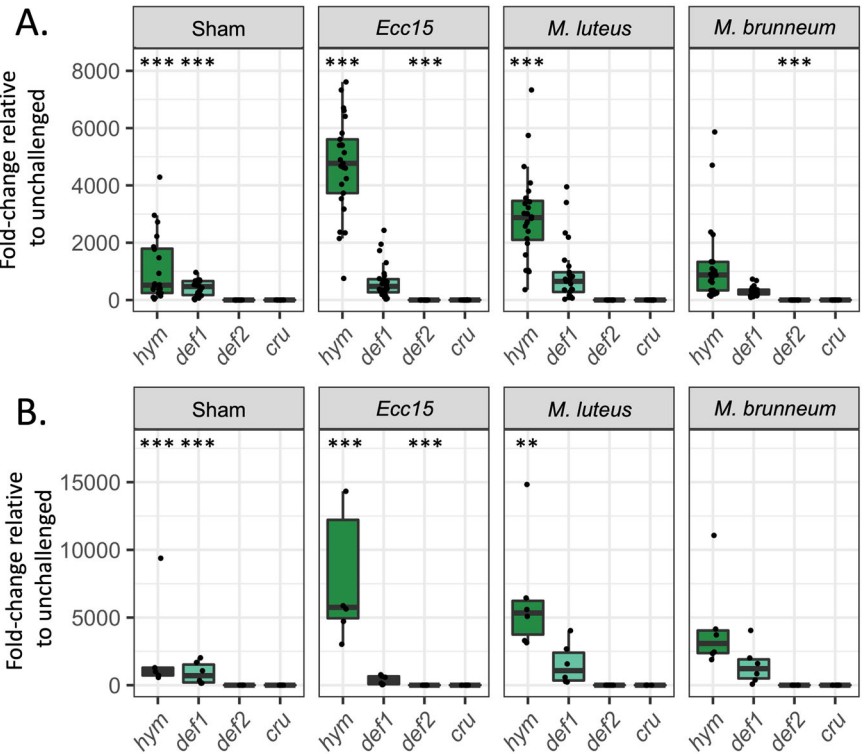

**Fig. 2 | AMP gene expression upon individual immune challenge.** RT-qPCR measurement of the induction of AMP coding genes in response to systemic infections in workers (**A**) and in queens (**B**), expressed as a fold-change of expression compared to unchallenged individuals ($N = 20–26$). Dots represent individual data points. Boxplots represent the median and interquartile range of biological replicates. The upper/lower whiskers are the values within 1.5 times the interquartile range over/under the 75th percentile. Data were analyzed by Linear Mixed Model (LMM) followed by post-hoc contrasts with Benjamini-Hochberg correction for multiple comparisons. Asterisks in the sham box indicate a significant difference compared to unchallenged. Asterisks in the pathogen boxes indicate a significant difference compared to the sham. ***$p < 0.001$; **$p < 0.01$. Significant differences imply an upregulation for *hym* and *def1* but a down-regulation for *def2*. Source data are provided as a Source Data file.

unaffected by challenges, and thus unlikely to be major effectors of the systemic immune response.

Remarkably, *hym*, *def1* and *cru* were not induced by fungal challenge (Fig. 2A; *M. brunneum* vs sham: $|z| < 1.061$, $p > 0.288$ for all comparisons), and *def2* was significantly downregulated by spore injection ($z = 3.80$; $p < 0.0001$), suggesting that *L. niger* NF-κB pathways do not respond to systemic fungal infections by producing AMPs.

As the lack of AMP response upon fungal infections was unexpected, we sought to confirm this with additional experiments. In nature, fungal contamination is mediated by conidiospores, an external, infectious form able to cross the insect cuticle, while proliferation within the insect body involves a different form called blastospores[33]. To test whether the lack of AMP induction was due to injecting conidiospores rather than blastospores, we compared the response to the two forms of the fungus. There was no difference in *hym*, *def1* or *cru* expression between sham, conidiospore- or blastospore-injected ants, which rules out any form-specific detection defect (Supplementary Fig. 3A). As in the previous experiment, *def2* was downregulated by conidiospores, and even further by blastospores.

Several species of *Metarhizium* perform immune evasion when infecting insects through the secretion of a protein coat that masks their glucan immunogenic structures[34]. To measure AMP responses in the absence of active evasion mechanisms, we denatured proteins with a heat treatment before injecting the spores (Supplementary Fig. 3B). We observed no induction of *hym*, *def1* or *cru* in individuals that received live or heat-killed spores compared to sham-treated individuals, and only mild induction of *def2* after injection of heat-killed spores but not live spores. These results overall indicate that the lack of AMP induction after live spore injections is not due to an active evasion mechanism from the pathogen but rather reflects a lack of host response.

Finally, injecting heat-killed spores of *Beauveria bassiana*, another entomopathogenic fungus, led to a comparable AMP expression profile to that observed after heat-killed *M. brunneum* injection (Supplementary Fig. 3C). This indicates that the lack of fungus-specific response is a general feature of *L. niger* individual immunity, rather than specific to *M. brunneum*.

We repeated the same experiment using queens, which displayed a qualitatively similar immune ability to workers (Fig. 2B) indicating that the ability of an individual to mount an immune response upon microbial challenge is not caste-dependent in *L. niger*. As reproductive individuals have a different physiology than that of workers, we also measured AMP expression in the absence of any immune challenge in virgin, claustral (mated but isolated) and colonial (surrounded by workers) queens, and in males (Supplementary Fig. 4). The expression of *def1*, *def2*, and *cru* differed significantly across groups, whilst the expression of *hym* did not (LMM, *hym*: $\chi^2 = 6.67$, df = 3, $p = 0.0834$; *def1*, *def2* and *cru*: $\chi^2 \geq 14.49$, df = 3, $p \leq 0.0023$). In particular, *cru* expression changed across life stages, with the highest expression observed in virgins and males but the lowest in claustral queens (Supplementary Fig. 4). Collectively, these data suggest that reproductive individuals have a distinctive AMP gene expression profile independently of infection.

## Ants are susceptible to opportunistic bacterial infections

Our results indicate that *L. niger*'s immune system specifically responds to bacteria, but not to fungi. This raises questions about the selective pressures allowing the evolutionary maintenance of such a specialized immune system. Rampant bacterial infections in wild colonies are rarely

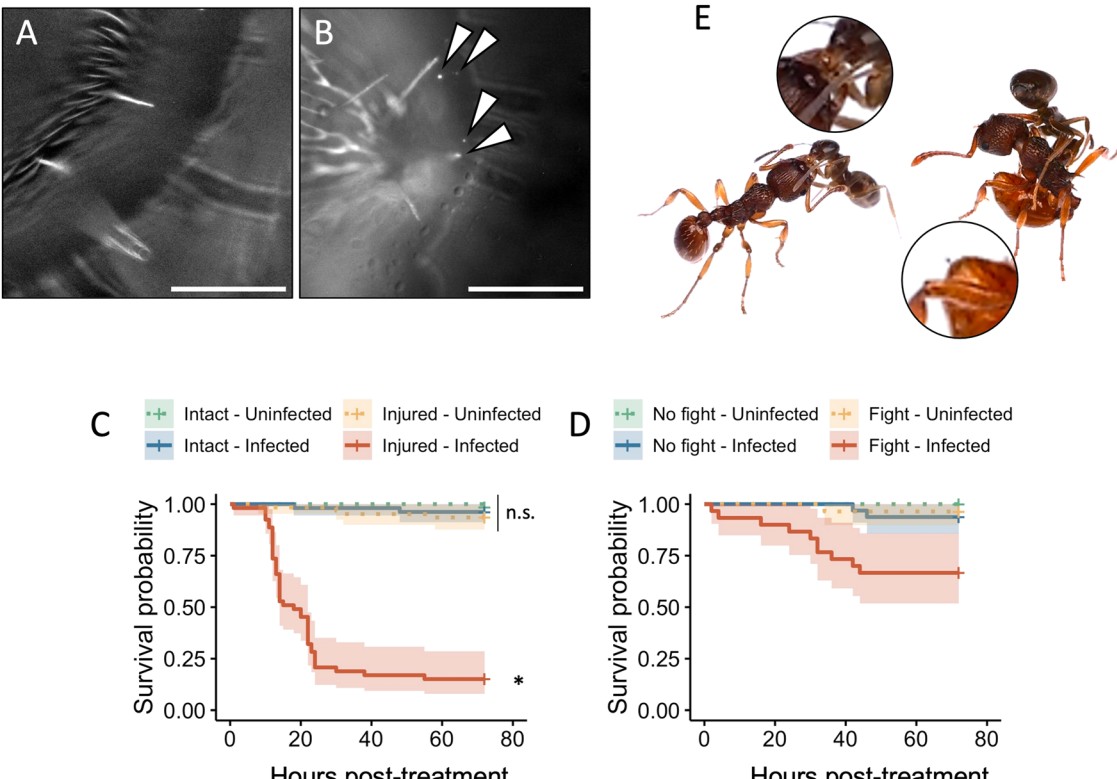

**Fig. 3 | Workers are susceptible to opportunistic bacterial infections. A–B** Thorax cuticle of a worker dipped (**A**) in a sham solution or (**B**) in a suspension of *P. entomophila*-GFP (arrowheads). Scale bar = 100 μm. *N* = 5. **C** Survival of *L. niger* workers as a function of body integrity (Intact vs. Injured) and exposure to a *P. entomophila* suspension (Uninfected vs. Infected). Shading indicates 95% confidence intervals. *N* = 30 per condition. Data were analyzed using a mixed-effects Cox model. 'n.s.' indicates no significant difference between treatments, whilst the asterisk indicates a statistically significant difference with all other treatments.

**D** Survival of *L. niger* workers as a function of fight with a *M. rubra* worker (No fight vs. Fight) and exposure to a *P. entomophila* suspension (Uninfected vs. Infected). Shading indicates 95% confidence intervals. *N* = 30 per condition. Data were analyzed by comparing the Akaike information criterion (AIC) of mixed-effect Cox models with or without interaction between fight status and exposure. **E** Illustrations of aggressive behaviors between *L. niger* and *M. rubra*. Left is a mandible grabbing. Right is *M. rubra* drawing its stinger. Source data are provided as a Source Data file.

observed[35], yet the cuticle of several ant species is known to frequently carry pathogenic bacteria[36,37]. Therefore, we hypothesized that a systemic antibacterial immune response could be ecologically relevant to fight opportunistic infections, that is, commensal bacteria becoming pathogenic upon gaining access to the inner tissues following a cuticle breach.

We tested this with the bacterium *Pseudomonas entomophila*, a soil-dwelling, generalist entomopathogen[38]. First, we verified that *P. entomophila* adheres to the cuticle of workers by dipping them briefly in a low-density suspension of fluorescent bacteria. Bacteria were visible on their thorax 10 min later, indicating that exposure in the wild can lead to bacterial presence on the cuticle (Fig. 3A, B). We then tested the consequence of exposure to *P. entomophila* following a sterile cut to one antenna (Fig. 3C). We observed no significant effect of sterile injury or exposure alone on ant survival, but most ants exposed to *P. entomophila* after being injured died within a day (median survival = 18 h; mixed-effects Cox Model (Coxme) with colony as random effect, interaction exposure×injury: Hazard Ratio (HR) = 15.30 ± 1.37, z = 2.04, *p* = 0.041; 'Exposed-Injured' versus every other condition: |z| > 4.93; *p* < 0.0001; all other pairwise contrasts: |z| < 1.26; *p* > 0.311), indicating that a wound is a fatal entry way for entomopathogenic bacteria.

We then assessed the ecological relevance of such opportunistic bacterial infections. Ants are generally aggressive with other species, so we hypothesized that interspecific fights could cause injuries which could be fatal if the ant cuticle harbored entomopathogenic bacteria. We tested this by setting up pairwise encounters between workers of *L. niger* and *Myrmica rubra*, a sympatric species, then exposing them to *P.*

*entomophila* and monitoring their survival (Supplementary Movie 1). During encounters (hereafter "fights"), we observed aggressive behaviors such as biting, which can cause microlesions especially on the joints, or *M. rubra* worker drawing its stinger which can cause puncture wounds (Fig. 3E). Though only one fight resulted in visible injury (antennal loss), *L. niger* mortality was five to ten times greater when exposed to *P. entomophila* after a fight than after bacterial exposure alone or after fighting alone (Fig. 3D). As no ant died in the control (no fight – no exposure), it was not possible to apply standard Wald tests of significance to the survival analysis. However, including an interaction between fighting and *P. entomophila* exposure improved the model fit (Coxme models with replicate as random effect, Akaike information criterion; $AIC_{[Infection + Fight]}$ = 13.02; $AIC_{[Infection + Fight + Infection:Fight]}$ = 11.33, difference = 1.69; see "Methods"), confirming that ant survival to an external bacterial challenge is jeopardized by fighting. Although we cannot rule out that this may be due to combat-induced stress[39], it is more likely a consequence of fight-related cuticle lesions, as these results closely mirror those produced by experimenter-inflicted sterile injury (Fig. 3C). Together, these results suggest that ant workers are susceptible to opportunistic bacterial infections resulting from cuticle breaches, and that such lesions can occur in ecologically relevant contexts such as territorial fights.

## Individual and social immunity have complementary functions against different classes of pathogens

To confirm that systemic individual immunity is effective against bacteria but not fungi, we compared the antimicrobial activity of the

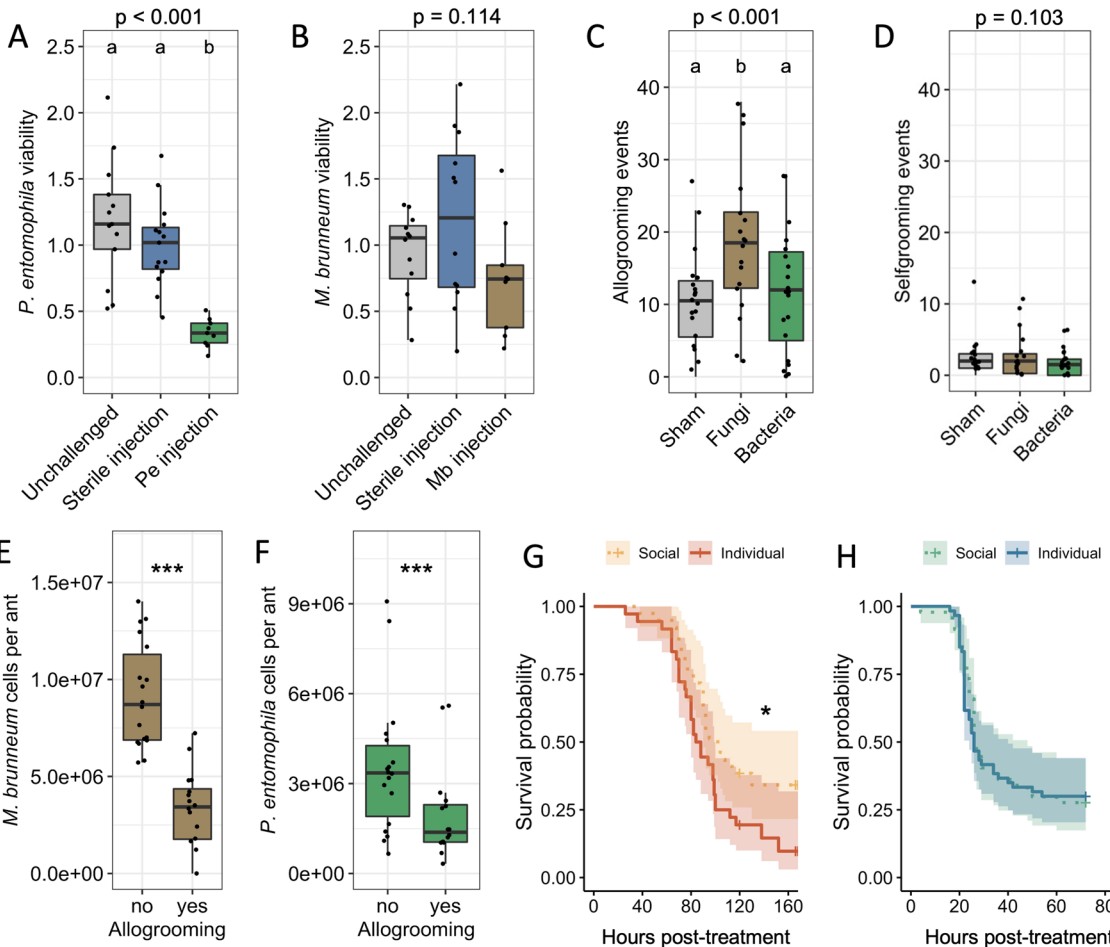

**Fig. 4 | Individual and social immunity confer complementary protection against different classes of pathogens.** Viability of (**A**) *P. entomophila* and (**B**) *M. brunneum* blastospores incubated in hemolymph extracted from unchallenged (grey), sterile-challenged (blue) or primed ants challenged with an injection of heat-killed microbes (*P. entomophila* 'Pe', green; *M. brunneum* 'Mb', brown; *N* = 9–15). Data were normalized to the mean of the sterile challenge control and analyzed using LMM. Number of (**C**) allogrooming and (**D**) selfgrooming events observed after external exposure of a focal ant to sterile buffer (sham control), fungal spores or bacteria (*N* = 18–20 per condition). Data were analyzed by Generalized LMM (GLMM) with Poisson distribution. qPCR quantification of (**E**) *M. brunneum* and (**F**) *P. entomophila* before and after allogrooming by nestmates (*N* = 16 per condition).

Data were analyzed by LMM. (**A**–**F**) Dots represent individual data points. Boxplots represent the median and interquartile range of biological replicates. The upper/lower whiskers are the values within 1.5 times the interquartile range over/under the 75th percentile. (**A**–**D**) P-values for the main treatment effect are indicated on top of each plot. Letters identify groups that were statistically different in post-hoc contrasts with Benjamini-Hochberg correction for multiple comparisons. (**E**–**F**) ****p* < 0.001 (LMM). Survival of injured ants exposed to (**G**) *M. brunneum* or (**H**) *P. entomophila* when kept socially or in isolation (*N* = 60 for **G** and *N* = 40 for **H**. Shading indicates 95% confidence intervals. Data were analyzed by using a mixed-effect Cox model. **p* < 0.05. Source data are provided as a Source Data file.

hemolymph between unchallenged ants and ants challenged with a sterile injection or primed with an injection of heat-killed bacteria or fungi, expected to elicit a specific immune response as described in Fig. 2. Hemolymph was treated to eliminate immune cells and prevent melanization, thereby ensuring that AMPs were the main immune effectors in play. Our results show that priming the ants with heat-killed *P. entomophila* significantly increased their hemolymph activity against the same microbe, whilst a sterile injection did not (Fig. 4A; LMM, effect of treatment, $\chi^2$ = 48.99, df = 2, *p* < 0.0001; sterile injection vs unchallenged: z = −1.23, *p* = 0.218; sterile injection vs *P. entomophila* priming: z = −5.75, *p* < 0.0001). By contrast, neither sterile injection nor priming the ants with heat-killed *M. brunneum* increased their hemolymph activity against the same microbe (Fig. 4B; LMM, effect of treatment, $\chi^2$ = 4.34, df = 2, *p* = 0.114). This confirms that internal bacterial challenge triggers systemic antibacterial immunity, but internal fungal challenge does not trigger systemic antifungal immunity.

Conversely, to test that social immunity is more effective against fungal than bacterial infections, we focused on allogrooming, a major social immunity mechanism that consists in the removal of infectious

particles from the cuticle of an ant by nestmates[3,40]. We found that external exposure to solvent (sham), bacteria or fungi elicited significantly different levels of allogrooming (Fig. 4C; LMM $\chi^2$ = 51.8, df = 2, *p* < 0.0001), but not selfgrooming (Fig. 4D; LMM $\chi^2$ = 4.6, df = 2, *p* = 0.103). More specifically, exposure to fungal spores induced significantly more allogrooming (18.6 events/focal ant) than either sham (10.4 events/focal ant) or bacterial exposure (11.7 events/focal ant; fungus vs sham: z = 6.55, *p* < 0.0001; fungus vs bacteria: z = 5.41, *p* < 0.0001; sham vs bacteria: z = 1.24, *p* = 0.21). We then measured the efficiency of allogrooming at removing microbes from the ants' cuticle. The load of both microbes was significantly reduced on focal ants after grooming, but to a greater extent for fungi (Fig. 4E; 62% reduction after grooming; LMM $\chi^2$ = 62.25, df = 1, *p* < 0.0001) than for bacteria (Fig. 4F; 41% reduction after grooming; LMM $\chi^2$ = 12.94, df = 1, *p* = 0.0003). Overall, these results indicate that allogrooming is not increased by bacterial exposure, and that basal allogrooming levels (similar between sham- and bacteria-exposed ants) are less efficient at decreasing microbial load than the induced allogrooming levels elicited by fungi.

Our hypothesis that social immunity is effective against fungal, but not bacterial infections implies that group living should provide a survival advantage to ants contaminated with fungi, but not bacteria. We tested this by comparing the survival of ants kept in groups or in isolation after being exposed to *M. brunneum* spores or injured and exposed to *P. entomophila*. As predicted, ants kept in groups survived significantly better when exposed to fungi, but not following opportunistic bacterial infections (Fig. 4G, H; median survival in group vs in isolation, *M. brunneum*: 102 h vs 86 h; Coxme: z = −2.17, *p* = 0.03; *P. entomophila*: 26 h vs 27 h; z = −0.26, *p* = 0.79). We tested the generalizability of this result by repeating the experiment with another live fungus, *B. bassiana*, and another bacterium, *Ecc15*, and found similar results, with ants kept in groups surviving better upon challenge with *B. bassiana* (Fig. S5A; median survival in group vs in isolation: 132 h vs 113 h; z = −1.97, *p* = 0.048) but not upon challenge with *Ecc15* (Fig. S5B; < 50% dead for both treatments; z = −1.45, *p* = 0.148).

In summary, these results support our hypothesis that individual and social immunity are complementary in fighting different classes of pathogens. Individual immunity is specifically elicited and efficient against bacterial opportunistic infections, whereas social immunity is inefficient against the latter but confers protection against fungal infections.

## Discussion

The genomic erosion of NF-kB pathways in social insect genomes has raised considerable debate about their functionality. Here, we confirmed that the Toll and the IMD pathway of *L. niger* are depauperate compared than those of solitary species, especially with regards to regulatory and effector genes, as previously shown for other social insects[9,10,15,16] (Fig. 1 and Supplementary Tables 1 and 2). In addition, we uncovered a reduced number of GNBP coding genes, seemingly specific to *L. niger* and at least two other *Lasius* species, which suggests an impaired Toll pathway activation by fungi in this genus. Analysis of AMP expression profiles upon pathogen challenge confirmed that the main systemic immune response in *L. niger* lies in a bacteria-specific activation of the IMD pathway, leading to the induction of *hym* only. This is consistent with previous findings that *hym* is also the only AMP upregulated by bacteria injection in queens of the dolichoderine ant *Linepithema humile*[41], suggesting a conserved function of *hym* as the main antibacterial AMP across ant subfamilies.

We observed similar levels of *hym* induction by two bacteria with different Gram staining (Fig. 2), which contrasts with typical expectations that Gram-negative and Gram-positive bacteria should specifically elicit the IMD and the Toll pathway, respectively[14]. Indiscriminate systemic immune responses have been reported in several insects and other arthropods harboring incomplete signaling cascades[42,43]. For example, in the hemipteran stinkbug *Plautia stali*, both Gram-negative and Gram-positive bacteria elicit the same immune genes, indicating that the functional differentiation of the two pathways is abolished[42]. In *L. niger*, the genomic conservation of IMD and Toll pathway core genes combined with the lack of discrimination between different types of bacteria suggests a similar cross-talk between pathways. More specifically, the presence of both Dorsal and Relish predicted binding sites within the promoter region of *hym* (Supplementary Fig. 2) and the broad spectrum of activity of Hym against Gram-negative and Gram-positive bacteria[19] suggest that *L. niger* may have evolved a minimalist immune system where both NF-kB pathways converge towards the induction of a single 'Swiss army knife' AMP. This would minimize the investment in individual immunity in favor of other biological processes while maintaining a minimal but functional line of defence against most bacterial infections.

This hypothesis raises questions about the selective pressures leading to the conservation of the whole NF-kB pathways, as opposed to incomplete pathways as found in some other organisms[42]. In *D. melanogaster*, both the Toll and the Imd pathway are also involved in antiviral defenses[22,44]. Viral infections are common in ants[45], indicating that the conservation of core genes of both pathways in *L. niger* could be due to viral pressure. Furthermore, pathway selection could be related to non-immune functions, such as apoptosis and autophagy for IMD[46–48], and embryonic development for Toll[49].

Whilst *hym* clearly seems to play a role in systemic immunity, the expression profile of *def1* and *def2* is more puzzling, as *def1* was induced by injury but not by infections, and *def2* was repressed upon Gram-negative and fungal infections. Although some insect Defensins have a direct antimicrobial activity[50], this family of AMPs carries out broader functions. For example, a honeybee Defensin promotes wound healing and tissue repair[23], and mammals' β-defensin-3 facilitates wound healing through the elicitation of the JAK-STAT pathway[51]. Therefore, Def1 and Def2 are interesting candidates in the regulation of the JAK-STAT pathway in *L. niger*, but this remains to be demonstrated empirically.

*Crustin* expression remained unchanged across all experimental challenges. Members of this family have diverse functions beyond direct antimicrobial activity and can be elicited by viral challenges and by heat or osmotic shocks[25]. It is therefore possible that we did not apply the appropriate stimulus to activate *crustin*, or that Cru is a venom component rather than an inducible antimicrobial, as is suspected for Waprin[27]. Alternatively, Cru could act as a prophylactic AMP in virgin queens and males, where its constitutive expression was highest (Supplementary Fig. 4), suggesting that it could be secreted by reproductive epithelia.

Surprisingly, challenge with the fungus *M. brunneum* did not induce any of the four *L. niger* AMP coding genes (Fig. 2), although it strongly elicits AMP gene expression in other non-Hymenoptera insects[52]. This apparent lack of a functional systemic immune response against fungi is further supported by the lack of one GNBP orthogroup in the genome, by the pseudogenization of one of the copies in the other orthogroup, and by the fact that internal fungal pathogen challenge did not increase the antifungal activity of the ant hemolymph, whilst internal bacterial pathogen increased hemolmpyh antibacterial activity (Fig. 4A–B). The absence of an immune response against fungi contrasts with previous studies reporting the induction of immune genes following natural infections in related ant species (*i.e.* upon external exposure rather than systemic injection). For example, *Lasius neglectus* workers increase their *defensin* expression ca. 5-fold after social contact with fungus-contaminated nestmates[11]. This increase is, however, dramatically lower than *def1* induction after injection of bacteria in *L. niger* (ranging between 200-fold and 600-fold compared to unchallenged, Fig. 2) and to that observed upon similar infections in other insects[21]. As we found similar induction of *def1* after sterile injury and fungus injection (Fig. 2), it is likely that the mild elicitation of *defensin* after external fungal infections is due to cuticle breach and epithelium disruption occurring during the infection process of *M. brunneum*[33], and hence reflects a generic tissue-damage response rather than a fungus-specific immune response. Our results, however, do not exclude the involvement of other individual immunity mechanisms active against fungi, such as melanization or cellular immunity[14].

In contrast to individual systemic immunity, which seems geared towards fighting bacterial infections rather than fungal infections, our results indicate that social immunity, whilst effective against fungal pathogens[1], provides little to no protection against bacterial infections in *L. niger*. Indeed, the presence of bacteria on the ant cuticle did not elicit increased allogrooming, and the presence of nestmates did not increase survival upon opportunistic bacterial infections (Fig. 4C and Fig. 4H). Together, our data thus indicate that individual and social immunity carry out non-overlapping, complementary defense functions against different classes of pathogens. We propose that the mode of contamination of entomopathogenic fungi (external exposure followed by active penetration of the cuticle to reach inner tissues after

24 to 48 hours[33]) provided a unique opportunity for the evolution of social care mechanisms specifically aiming at removing or killing spores before they penetrate the insects' cuticle. The effectiveness of these social care mechanisms is likely to have relaxed the selection pressures for the maintenance of an effective individual immune response against fungi, leading to the erosion of individual fungal-specific immune defenses. By contrast, our results indicate that external exposure to bacteria is not a threat to *L. niger* ants if their cuticle remains intact. As *L. niger* mostly feeds on insect remains, nectar and honeydew, we assume that natural injuries are rare in this species and that effective social care mechanisms such as bacterial detection, removal and disinfection may not have been selected over *L. niger* evolution. This would have maintained the need for an effective individual immune response to fight occasional injury-mediated opportunistic infections. Formal testing of this hypothesis would require extended comparative studies, but indirect support can be found in the termite-hunting ponerine ant *Megaponera analis*. Because this ant feeds exclusively by raiding termite mounds, the workers are subject to high rates of injuries, which can get infected by pathogenic soil bacteria, greatly increasing mortality[53,54]. Interestingly, this species has evolved a social defense mechanism that significantly decreases the risk associated with injury-mediated opportunistic bacterial infections: injured workers receive wound care that includes wound licking and deposition of antimicrobial secretions[53,55]. Comparisons between *L. niger* and *M. analis* support the hypothesis that the likelihood of injuries may influence selection pressures for the evolution of bacteria-targeting social immune mechanisms, although further comparative studies will be required to test this hypothesis.

In summary, our work highlights the complexity and complementary functions of immune defenses in ants. With over 15,000 described species, ants are one of the most ubiquitous and ecologically diverse insect groups on Earth. Social immunity mechanisms vary dramatically between species and across social contexts[6,56]. Social insects' individual immunity mechanisms, on the other hand, are often assumed to mimick mechanisms described in non-social model species such as *D. melanogaster* and their functionality often lacks empirical validation. We argue that further research should focus on the functions of social insects' immune system in a more standardized way and in a broader range of species and contexts. Such efforts are essential to refine our understanding on how the interplay between individual and social immunity provides the most cost-effective defense against pathogens in light of a species' life history traits.

## Methods
### Genome assembly
The *Lasius niger* colony used for genome sequencing was collected in 2018 in Belgium by Serge Aron as part of the Global Ant Genomics Alliance (GAGA) initiative[57]. DNA was extracted from ca. 30 workers using phenol/chloroform extraction and used to construct a PacBio Continuous Long Read (CLR) library with an average size of 30 kb using SMRTbell Template Prep Kits according to the manufacturer's instructions. The library was sequenced on a PacBio Sequel platform at Novogene (Tianjing, China). Single-tube long fragment read (stLFR) libraries were prepared and sequenced by the Beijing Genomics Institute (BGI) using the BGISEQ-500 platform. In addition, a Hi-C library was constructed using 4 pupae without cocoon following published protocols[58]. The Hi-C library was sequenced on a BGI DNBseq platform with 100 bp paired-end mode.

The PacBio reads were cleaned using the PacBio SMRT-Analysis package, removing sequencing adapters and filtering reads with low quality and short length (parameters: minSubReadLength:500). The clean reads were then assembled using Wtdbg2 v2.5 (-t 16 -x sq -g 300 m) and scaffolded using SSPACE-LongRead. An additional round of gap filling was conducted using LR_Gapcloser (https://github.com/

CAFS-bioinformatics/LR_Gapcloser) with the PacBio subreads. A first round of polishing was performed using Arrow (https://github.com/skoren/ArrowGrid) to map the PacBio sequences to the genome assembly and correct the small indels and substitutions in the initial assembly. An additional polishing step was performed using the short stLFR reads cleaned from adaptors and PCR duplicates using stlfr2supernova_pipeline (https://github.com/BGI-Qingdao/stlfr2supernova_pipeline). The consensus genome sequences were then polished with the clean stLFR reads using NextPolish v1.3.0. Finally, the polished scaffolds were further scaffolded using the barcoding information from stLFR reads with SLR-superscaffolder (https://github.com/BGI-Qingdao/SLR-superscaffolder). Duplicate scaffolds were identified and filtered using Funannotate "clean" pipeline v1.8.3 (https://github.com/nextgenusfs/funannotate). The de novo *Lasius niger* PacBio-based genome assembly and the Hi-C library reads were used as input data in the 3D-DNA pipeline[59] to generate chromosome length scaffolds. The quality of the assembly was assessed by visualizing the Hi-C contact map and manually curated for the final assembly using JuiceBox (Supplementary Fig. 6).

The final chromosome level assembly was screened for putative contaminations by dividing in 2000 bp sliding windows with 500 bp overlap and searching each window against insect and bacterial databases using mmseqs (release_12-113e3). The ratio of windows showing similarity to bacterial or eukaryotic sequences was used, along with the coverage and GC content, as evidence to identify contaminant scaffolds. Assembly quality was evaluated using contiguity metrics, gene completeness with BUSCO v5.1.2[60] and compleasm v0.2.2[61], and consensus quality (QV) and k-mer completeness using Merqury[62] (Supplementary Table 4). Finally, mitochondrial (CO1 and CytB) and autosomal (Wingless, LwRh, AbdA, ArgK) markers were annotated in the assembly using BITACORA[63] to confirm the species identity.

Homology-based and de novo methods were conducted to identify transposable elements (TEs). The genome sequences were aligned against the Rebase TE library (v25.03) and TE protein database using RepeatMasker and RepeatProteinMask (version 4.1.2)[64]. In addition, RepeatModeler v2.0.2[65] was used to build de novo *L. niger* repeat library, which was subsequently used to annotate repeats using RepeatMasker. TRF v4.10.0[66] with Match = 2, Mismatch = 7, Delta = 7, PM = 80, PI = 10, Minscore = 50. Finally, we combined all evidence resulting in 33.44% of the assembled genome being repeat sequences, for a total length of 96 Mb (Supplementary Table 4).

### Genome annotation
RNA-seq from different *L. niger* castes generated by the GAGA project were aligned to the reference repeat soft-masked assembly using STAR v2.7.2b default options[67]. In addition, the annotations from the publicly available genome of *D. melanogaster*, *Tribolium castaneum*, *Nasonia vitripennis*, *Apis mellifera*, *Ooceraea biroi* and *Camponotus floridanus* were used to conduct homology-based gene predictions using GeMoMa v1.7.1, which also incorporates RNA-seq evidence for splice site prediction[68]. The independent RNA-seq alignments were then merged creating a consensus Gene Transfer Format using Stringtie v2.1.5[69], and BestORF (Molquest package, Softberry) was used to identify open reading frames (ORFs) in the transcript sequences. Transcripts with incomplete ORFs were filtered out. We randomly selected ~1000 high–quality genes from GeMoMa prediction to train Augustus v3.2.2[70]. The de novo gene prediction was then performed using Augustus with the repeat-masked genome, filtering out genes with lower length than 150 bp or incomplete ORFs. Finally, gene annotations from all evidences were combined, generating the final gene annotation including an annotation with a single representative isoform (longest) per gene. Transposon-related proteins were identified and filtered using a Basic Local Alignment Search Tool (BLASTp)

search against Swissprot database and the transposable element protein database from RepeatMasker.

## Immunity gene annotation

A custom database of *D. melanogaster* genes associated with the Toll and IMD pathways was built using the Pathway Report Lists from the Flybase database[71]. The list was manually reviewed to remove duplicates, as well as genes that were only distantly related to the considered pathway based on published evidence, leaving a total of 152 unique queries. All predicted polypeptides for each gene were then extracted using Flybase Batch Download tool to build the query database. Homologues of each query sequence were searched against the *L. niger* predicted proteome using local BLASTp on Geneious 2021.2.2 (https://www.geneious.com) (BLOSUM62, word size = 3, gap penalty opening = 11 and extension = 1). A homologue was identified upon reciprocal best BLASTp hit between the *D. melanogaster* and *L. niger* sequences with e-value < $10^{-20}$. An extra step was taken for genes initially presumed absent when they fulfilled at least one of two conditions: (1) homologues of genes known to functionally interact with them were identified in the *L. niger* genome, or (2) homologues were previously annotated in other Hymenoptera species. In such cases, a homologue of the *D. melanogaster* query was first identified in *A. mellifera*, and the *A. mellifera* homologue was then blasted against the *L. niger* proteome using the same workflow as for direct *D. melanogaster – L. niger* searches.

We further explored annotated *Spz* and *Toll* genes to assess the gene family conservation across Formicidae, Apoidea, *D. melanogaster* and *T. castaneum*. Specifically, we used the orthology assessment generated using Orthofinder v2.5.4[72] across 163 ant genomes sequenced by the GAGA project (https://db.cngb.org/antbase/project), and the abovementioned outgroup species. We selected four representative species: *L. niger, Monomorium pharaonis, A. mellifera* and *D. melanogaster*; and aligned the protein sequences of the genes assigned in the *Spz* and *Toll* orthogroups, separately, using MAFFT-linsi accurate mode v7.453[73]. We used IQTREE v2.1.3[74] to generate gene phylogenies using ModelFinder and retrieving branch supports from 1000 Ultrafast Bootstrap replicates.

Clear identifications of *SPE* and *Grass* were made difficult by the high similarity across the serine protease family. Therefore, their presence in *L. niger* genome was considered as 'uncertain'.

For AMP annotation, we extended our query database with the full APD3 Antimicrobial peptide sequences 2020 release containing 3167 AMP sequences from all kingdoms of life[75]. 76 queries had a significant BLASTp hit against *L. niger* predicted proteome with an e-value cut-off of $10^{-5}$ and matched 33 unique *L. niger* candidate peptides. Candidates were subsequently blasted against the nr (non-redundant) database to remove false positives and searched on InterProScan to verify the presence of conserved domains from their respective AMP family.

## GNBP orthology analysis

GNBP orthology was analyzed using Orthofinder v2.5.4 across 163 ant genomes from the GAGA project or Genbank database, 8 Apoidea species and additional 5 Hymenopera with high quality genome and annotations, and *Tribolium castaneum* and *Drosophila melanogaster* as outgroups. Phylogeny was reconstructed using a subset of sequences for 17 ant genomes, 8 additional Hymenoptera and the two outgroups (list in Fig. S1C). MAFFT L-ins-I v7.453 was used to perform the protein alignments, which were filtered using trimal (-gappyout option). IQTREE v2.1.3 was used to reconstruct the maximum likelihood gene tree with 1000 ultra-fast bootstrap replicates.

## Prediction of NF-κB binding sites

As consensus NF-κB binding sites are unknown in *L. niger*, we took advantage of their high level of conservation across species to search for *D. melanogaster* consensus binding sites on the *L. niger* genome.

The Position Weighted Matrices (PWMs) of the consensus binding sites of the *D. melanogaster* proteins Relish and Dorsal were retrieved from the OnTheFly database[76]. PWMs were used as input on the Find Individual Motif Occurrences (FIMO) software[77] to scan both strands of the 500 bp upstream of each AMP coding sequence. Site prediction was deemed significant when it reached a *p*-value < 0.001.

## Insect collection and husbandry

Newly mated *L. niger* queens were collected in Bristol, UK, in July 2021 and 2022 and allowed to found new colonies in the lab. Colonies were reared in fluoned boxes at 25 °C with 65% humidity and 12 h/12 h light/dark cycle, and were provided with water, 15% golden syrup solution and frozen *Drosophila hydei*. Colonies were left to grow for a year and each contained at least 50 workers before ants were sampled for experiments. *Myrmica rubra* colonies were collected from Langford, UK, in July 2022 and kept in the same conditions with weekly provision of frozen Pacific cockroaches instead of *Drosophila*. Virgin *L. niger* gynes and males were frozen immediately after capture at the exit of a colony during a mating flight in 2021. Claustral queens were captured after a mating flight, kept individually in the dark in glass tubes with water and without food, and frozen 3 weeks after collection.

## Experimental infections for gene expression assays

*Erwinia carotovora carotovora* (strain *Ecc15*[28]) and *Micrococcus luteus* (undeposited strain from the Lemaitre lab[78]) were cultured in lysogeny broth (LB) at 29 °C with 250 rpm shaking. Density was adjusted with LB to an Optical Density at 600 nm ($OD_{600}$) of 5 corresponding to ~$10^9$ cells/mL. *Metarhizium brunneum* (strain MA275 KVL-03-143) and *Beauveria bassiana* (strain 802[78]) were grown on Sabouraud-Dextrose-Agar (SDA) plates until sporulation was observed. Conidiospores were collected, washed and adjusted to a density of $10^9$ cells/mL in 0.05% Triton X-100. Heat-Killed conidiospores were obtained by heating the density-adjusted suspension at 98 °C for 1 h in sealed tubes. Heat-killed spores elicit the Toll pathway in a similar fashion to live spores in *D. melanogaster*[79]. Spore death was assessed by plating 50 µL of suspension on SDA plates and confirming the absence of colony growth after 48 h at 24 °C. Blastospores were obtained by inoculating conidiospores in Adamek's medium with minor modifications (4% glucose, 4% yeast extract, 3% cornsteep liquor, 0.4% Triton X-100, pH 6.8) and incubating the culture for 3 days at 24 °C and 150 rpm shaking. Blastospores were washed and adjusted to $10^9$ cells/mL in 0.05% Triton X-100.

23 nL of microbe suspension or spore suspension were injected in the gaster of worker or queen ants between the first and the second abdominal tergites using a Nanoject II (Drummond #3-000-206 A). Control injections ('sham treatment') were performed using 23 nL of LB (bacteria control) or 0.05% Triton X-100 (fungus control). Ants were kept in groups of 20–30 in sealed Petri dishes with water and sugar water at 24 °C for 24 h. Three biological replicates were performed on different colonies with 7–13 individual workers per condition for each replicate. Queens received a dose of 69 nL to account for their larger size and were incubated individually. At least five biological replicates containing a single queen were performed for each condition.

## RNA extraction and reverse transcription

Tissues were frozen at −80 °C in collection microtubes (Qiagen #19560) until RNA extractions. Each tube received 100 µL of Trizol (Invitrogen #15596026) and a single 2 mm pH-treated glass bead and was homogenized 2 × 2 min in a TissueLyzer II (Qiagen) at 30 Hz. Each tube then received 400 µL of Trizol and 150 µl of chloroform, was shaken strongly and left at ambient temperature for 5 min, then centrifuged at 2500 g for 20 min at 4 °C. The aqueous phase was transferred to a new tube containing 250 µL of isopropanol. Tubes were gently shaken, left for 5 min at ambient temperature, and at −80 °C for 1 h. Tubes were then centrifuged at 2'500 g for 40 min at 4 °C. The pellet was washed twice

with 400 μL ice-cold ethanol 70%, air-dried and resuspended in RNAse-free water. RNA concentration was adjusted to 250 ng/μL. 6 μL were submitted to a DNAse treatment in a 10 μL total volume using the DNA-free DNA Removal Kit (Invitrogen #AM1906), following manufacturer's instructions. 6 μL (900 ng) of DNA-free RNA was reverse-transcribed for 45 min using the PrimeScript RT Kit (Takara #RR037B) with an equal mix of oligo-dT and random hexamers. cDNAs were diluted 1/20 in mQ water prior to qPCR.

## Quantitative PCR for gene expression measurements

qPCR was performed on 2 μL of diluted cDNA mixed with 5 μL of PowerUp SYBR Green 2X mastermix (Thermofisher #A25742), 0.5 μL of reverse and forward primer 10 μM and 2 μL of mQ water. Primers sequences were: EF1a-F: 5′-ACGTCCAAATATTATGTCACCATCATC-3′; EF1a-R: 5′- CTTGCGAAGTACCAGTGATCATG-3′; Hym-F: 5′- AGCAACT-GACCACGGATCTG-3′; Hym-R: 5′- CGTAGACATCCGCCGTTGT-3′; Def1-F: 5′-GAGTACATCGGTGCCTCTGG-3′; Def1-R: 5′-CCTCTGA-GAAGGCAGTGAGC-3′; Def2-F: 5′-AGCTGTTTATGACGGGCCTAC-3′; Def2-R: 5′-CAGGAGAAAAGGTCGCAGGT-3′; Cru-F: 5′-CCTCATCGGAG-CATAGCTGG-3′; Cru-R: 5′-TAGGCGGCTTGTACCATTCG-3′. Cycling was performed with Tm = 60 °C following PowerUp SYBR Green mastermix instruction. The expression of the target gene was normalized to that of the housekeeping gene *EF1a* using the delta-CT (ΔCT) method and, for graphical display, each condition was normalized to its corresponding unchallenged control using the ΔΔCT method[80].

## PeGFP imaging

GFP-tagged *P. entomophila* L48 (Pe-GFP)[38] was cultured in LB overnight at 29 °C with 250 rpm shaking. Culture density was adjusted with PBS Triton-X100 0.005% (PBSt) to an $OD_{600}$ of 5. Single workers were dipped in PBSt or bacteria suspension for 5 sec and left individually in a dish for 10 min. Ants were freeze-killed and their thorax was observed on a Leica DM IRE2 microscope with a bandpass of 450–490 nm for excitation and 500–550 nm for emission. Exposure and gain were identical for controls and bacteria-exposed ants and contrast was adjusted on ImageJ to 0.10% saturated pixels.

## Experimental infections for survival assays

Infections for survival assays were carried out with *P. entomophila* adjusted to an $OD_{600}$ of 5 with PBSt and *Ecc15* adjusted to an $OD_{600}$ of 10 with PBSt. Sterile injuries were performed by cutting the right antenna scape of each worker with microscissors. *M. brunneum* and *B. bassiana* were prepared in the same way as for gene expression assays but resuspended in PBSt to -10⁹ cells/mL. Workers were dipped for 5 sec in the suspension or in sterile PBSt (sham) and left to dry on filter paper. Ants were kept at 24 °C, 60% relative humidity and constant low-intensity light in sealed Petri dishes with water and sugar water, either in groups of 10 ("social") or in isolation ("individual"). Death time was recorded from 5-minute videos acquired every 2 h with a Raspberry Pi computer with a camera module. 10 ants per condition were used in each biological replicate (different colonies), six biological replicates were performed for the Injury-Infection model (60 total ants/condition) and four replicates for the Social-Individual model (40 total ants/condition).

## Interspecific encounters

Workers from each species (*L. niger* and *M. rubra*) were enclosed individually in 1.5 mL tubes closed with a cotton ball. Within 5 min, each *M. rubra* worker was transferred into a tube containing a *L. niger* worker. 'No-fight' controls were made by enclosing two *L. niger* workers from the same colony. Immediately after transfer, the cotton ball was pushed into the tube to leave only -1 cm of free space, which initiated aggressive behaviors. Tubes were left undisturbed for 15 min, then the encounter was stopped by a pulse of $CO_2$. *L. niger* workers were immediately dipped into a *P. entomophila* suspension $OD_{600} = 5$

in PBSt or sterile PBSt, and their survival was monitored as described above. 6 ants out of 64 died during the encounters and were discarded. Three biological replicates (different colonies) were performed, each including 10 to 12 individual fights (30 to 36 total ants/condition).

Supplementary Movie 1 and Fig. 3E were acquired from encounters set up on a plastic bridge surrounded by water using an Olympus Lumix GH6 camera equipped with a 60 mm f/2.8 ED macro lens. Contrast and sharpness were enhanced on ImageJ.

## Antimicrobial assay

Hemolymph was extracted from claustral queens collected in Bristol in July 2023 and left undisturbed in water tubes for weeks at 24 °C in the dark. Extraction was done by piercing their pronotum using a Nanoject II and gently filling a glass capillary. Hemolymph was immediately transferred to a tube, centrifuged to eliminate debris and immune cells, and mixed with an equal volume of ice-cold Hemolymph Conservation Buffer (HCB; 20 mM Tris-HCl, 20 mM NaCl, 0.01% Triton X-100, 1% Phenylthiourea (PTU)). PTU inhibits melanization, thereby preventing interference of this mechanism during the assay. 24 h before hemolymph extraction, queens had received a thoracic injection of 32 nL of sterile PBSt ('Sterile injection') or 32 nL of heat-killed microbe suspension ('Priming'). Microbe suspensions were prepared at $OD_{600} = 5$ and heat-killed by 10 min (*P. entomophila*) or 1 h (*M. brunneum*) at 98 °C. Microbe death was assessed by plating 50 μL of suspension on LB or SDA plates and confirming the absence of colony growth after 48 h at 29 °C and 24 °C, respectively. Hemolymph samples were frozen at −80 °C as quickly as possible after dilution in HCB. 2 μL of diluted hemolymph were mixed with 2 μL of *P. entomophila* or 2 μL of *M. brunneum* blastospore suspension adjusted to 10⁵ cells/mL in HCB. Reactions were incubated for 3 h at 20 °C, then each reaction was diluted with 160 μL sterile HCB and 40 μL were immediately plated on LB plates with 50 μg/mL or rifampicin for *P. entomophila*, or Sabouraud-Dextrose Agar plates for *M. brunneum*. Plates were incubated for 24 h at 29 °C and 48 h at 25 °C, respectively, and Colony-Forming Units (CFU) were counted manually. Each biological replicate consisted of hemolymph extracted from 5 to 8 queens, for a total of 9 to 15 replicates per condition. Replicates were performed by series of 3 using the same microbe suspension. As colony number can vary significantly between series, counts for each condition were normalized to the mean CFU number of their matching sterile injection samples to calculate a relative viability index comparable across series.

## Allogrooming assay

-2 h before each assay, six focal (treated) ants were randomly selected in a colony, marked with a paint dot on their gaster and kept together in an empty Petri dish. At the start of each assay, three paint-marked focal ants were placed individually into a fluoned Petri dish arena (ø 50 mm, 20.3 mm height), with a plaster base (72 g fine plaster, 30 g dental plaster, 84 g water; 10 g per arena) along with 9 randomly selected unmarked nestmates and 5 brood items for a 30 min habituation period. Then, each focal ant was collected from the arenas and dipped for 5 sec into PBSt (sham), a *P. entomophila* suspension or a *M. brunneum* conidiospore suspension at 10⁹ cells/mL in PBSt. Suspensions were prepared following the procedure described above. Focals were dried on filter paper and left to recover individually for 30 min in separate dishes before being returned to their respective arena and recorded for 30 min using a 4 K Ultra-HD camera. Allogrooming (grooming of the focal ant by one nestmate) occurrence were annotated by a single observed for the first 20 min of recording using BORIS version 7.12.12[81]; the observer was blind to treatment at the time of the annotation. Twenty biological replicates (different colonies) were performed, each consisting in one focal ant per treatment. Two replicates were discarded because the focal ant did not move for the whole duration of the recording, or because it climbed on the dish wall away from the group.

## DNA extraction and quantitative PCR for quantification of microbial load

Focal ants were frozen individually at −80 °C at the end of allogrooming recordings. In parallel, control ungroomed samples were produced by exposing nestmates to the microbes and by freezing them immediately after the 30 min recovery period. Each tube received 50 μL of mQ water and a 2 mm glass bead and was homogenized in a TissueLyzer II for 2 × 2 min at 30 Hz. 180 μL of ATL lysis buffer (Qiagen #939011) and 20 μL of 20 mg/mL proteinase K (Qiagen #RP107B-1) solution were added and tubes were incubated at 56 °C overnight with 150 rpm shaking. Proteins were precipitated by addition of 160 μL of NaCl 5 M and centrifugation at 2500 g for 10 min. The supernatant was collected, mixed with an equal volume of 85% isopropanol and incubated at −20 °C for 2 h. Tubes were centrifuged for 30 min at 4'000 g and the DNA pellet was washed twice with 70% ethanol. Pellets were resuspended in 500 μL of mQ water. Microbes were quantified by qPCR by mixing 1 μL of DNA with 5 μL of PowerUp SYBRGreen 2X mastermix, 0.5 μL of reverse and forward primer at 10 μM, and 3 μL of mQ water. *M. brunneum*-specific primers and cycling conditions were published previously[5]. The following primers were used for *P. entomophila*: Monalysin-350F: 5′-GGCA-TACCCGTTCCTTCGAG-3′; Monalysin-439R: 5′-TGGAAATCTCCGAACC-CACG-3′ and cycling was performed following PowerUp SYBRGreen mastermix instructions with Tm = 60 °C. An absolute quantification was performed using a standard curve made from dilutions of purified genomic DNA of each microbe. DNA quantity per sample, expressed in ng/μL, was converted to number of microbial cells using a genome size of 38 Mb for *M. brunneum* and 5.9 Mb for *P. entomophila*, corresponding to 24 381 and 157 027 genome copies per ng of DNA, respectively.

## Statistical analysis

Data were analyzed on R v4.1.1 using RStudio v1.4.1717 and packages *lme4, coxme, multcomp, car*, and *survminer*. RT-qPCR, qPCR and antimicrobial assay data were analyzed using Linear Mixed-effects Models (LMM) with experimental condition(s) as fixed effect(s) and replicate and/or colony as random effect(s). Whenever necessary, data were transformed with a square or square root transformation to ensure normality of the residuals, which was validated using Shapiro-Wilks tests. Allogrooming and selfgrooming events were analyzed with General Linear Mixed-effect Models (GLMM) with Poisson distribution. Statistical significance of fixed effects was evaluated by analysis of variance based on Wald chi-square tests followed by post-hoc contrasts with Benjamini-Hochberg correction for multiple comparisons. Each point represents a biological replicate unless otherwise noted in the legend.

Survival data were analyzed using mixed-effects Cox models with experimental conditions as fixed effects and 'replicate' and/or 'colony' and/or 'experimentalist' as random effects. Statistical significance of fixed effects was evaluated by analysis of variance based on Wald chi-square tests followed by post-hoc contrasts with Benjamini-Hochberg correction for multiple comparisons.

No ant died in the control group of the fighting experiment (no fight−no infection), which causes the Cox coefficient to tend to infinity and makes the Wald test of significance unreliable. Instead, a likelihood-based method was applied to this dataset by comparing the Akaike Information Criterion (AIC) of a model without interaction term to that of a model with an interaction term and by deeming the model with the lowest AIC better.

## Reporting summary

Further information on research design is available in the Nature Portfolio Reporting Summary linked to this article.

## Data availability

The data generated in this study have been deposited in the Zenodo database under accession code 12820609. Numeric data generated in this study are provided in the Supplementary Information/Source Data file. The *L. niger* genome data is available on the NCBI BioProject Database under accession number PRJNA1159026. Additional genome-related data including the original assembly and annotations are available on the Electronic Research Data Archive at University of Copenhagen (https://doi.org/10.17894/ucph.1bca84a9-3b29-4628-90e5-097b4b11c413). Source data are provided with this paper.

## Code availability

Custom R code used to analyze and plot the data are available from the Zenodo repository (https://zenodo.org/records/12820609).

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

## Acknowledgements

We thank Prof Nicolai Vitt Meyling from the University of Copenhagen for providing the *M. brunneum* strain and Bruno Lemaitre from Ecole Polytechnique Fédérale de Lausanne for providing all other microbes. We are also grateful to Hannah Westlake for sharing her invaluable knowledge on *Drosophila* immunity, to Daniel Schläppi and to Paris Donelly for technical assistance, and to Luke Leckie, Beki Kennard and Tom Richardson for comments on the manuscript. FM, RB and NS acknowledge funding by the European Research Council (ERC Starting Grant 'DISEASE', no. 802628, to NS). RB and NS acknowledge funding by the Biotechnology and Biological Sciences Research Council (BBSRC grant BB/X511997/1, to the University of Bristol).

## Author contributions

F.M. and N.S. conceived the study. J.V. and Z.X. performed the genome assembly and annotation. F.M. performed the functional assays and immune gene annotation. R.B. and F.M. performed the allogrooming assay. T.I. performed microbe quantifications. J.R. and J.V. provided decisive input. N.S. acquired the funding. FM and NS wrote the original draft and all authors revised and approved the final manuscript.

## Competing interests

The authors declare no competing interests.
