## [Transparent Peer Review file · Nature Communications]

Pathogen-specific social immunity is associated with erosion of individual immune function in an ant

Corresponding Author: Dr Florent Masson

Version 0:

Reviewer comments:

Reviewer #1

(Remarks to the Author)

The manuscript by Masson is a very welcome test of a so far poorly resolved hypothesis of immune gene erosion in social insects, addressing a fundamentally important aspect of the relative role of social and individual immunity in an ant species. The manuscript sets out in an extensive and convincing set of experiments to show the immune gene repertoire in *Lasius niger* and testing specifically the effectors against fungal and bacterial infections in biologically-relevant contexts. The authors show that individual immunity protects against opportunistic bacterial infections which are not covered by social immunity. By contrast, social immunity targets fungal infections, which do not elicit an effective individual immune response. This provides an important next step in understanding how millions of years of evolution of optimised immune targets have resulted in complementary actions of social and individual immunity.

The experimental work present in the paper is solid, all conclusions warranted, and I have no major reservations of the experimental work. However, I feel that the study could benefit from better comparisons across ant species to allow assessing how generalisable these findings are and how they relate to ant ecology. I suspect that complementarity might well be old in the group and with many genomes available, it is possible to, at least at the genome level, ascertain this based on immune pathway annotations. I recognise that performing comparative experimental work across ant lineages will likely not be possible, but the molecular signals based on genomics could provide the authors the ability to better assess (and discuss) how general these patterns are predicted to be and potentially identify ecologically-relevant differences across species (e.g., related to whether species are predatory, herbivores, or generalists as is alluded to in the discussion but never explicitly tested).

Similarly, while both *Pseudomonas* and *Metarhizium* are good model pathogens for experimental tests, and vast comparisons across pathogens obviously limited, the authors should at the least discuss how generalisable the conclusions are likely to be.

Minor comments by line numbers:

Title: I would change "in ants" to "in an ant" or "in the common garden ant" to clarify that it is in one species and avoid overstating the findings.

Line 58-59: I recognise the need for brevity but some detail would be nice for readers not already familiar with the literature
Line 149 and elsewhere: giving the comparisons in parentheses is perhaps not needed as it's imperative that the represent a comparison of an experimental vs a sham treatment. Instead, I recommend giving the clear test in the text and then stating the type of test in parentheses

Line 162. Missing 'of' before AMP induction

Line 205: I think it should be 'and' instead of 'but'

Line 225. "hereafter" instead of "thereafter"?

Line 335: the authors come back to the point in the discussion that cuticle breaches may not be frequent in non-predatory ants, so a bit more caution here on the importance of breaches may be good to include. Has any work quantified the frequency and importance of breaches in nature?

Line 262: 'on' should be 'of'

Line 271: what is known about cue differences between fungi and bacteria? I also do not see a discussion of (hypothesis for) why social and individual immunity may be subject to complementarity in ants.

Line 275: I would cut 'Finally,'

Line 301: cut 'with'?

Line 352: why were these components not evaluated as well, what would it take, and what would the authors predict? While leaving these out does not take away from the clear patterns that the authors present, it prevents a more complete picture of the relative roles of (clearly important) immune mechanisms in ants.

The final paragraph of the discussion starting at Line 381 should be toned down to better represent the findings, avoid overstatements and present limitations. For example, stating that the work "...illustrates the diversity of immune defense mechanisms across this group, in the light of their ecology and evolutionary history limitations" based on a single ant species and two pathogens is a gross overstatement. It would be much more useful for readers if the authors focused on what this tells us, where the limitations are, and what incisive new questions with predictions are (e.g., patterns across the ant phylogeny).

Across all figure legends, specific method used (e.g., qPCR) should be mentioned as it is not clear from the figures themselves and sample sizes should be provided.

Figure S4 is not cited in the text as far as I can tell and it doesn't seem to be important for the paper.

I hope these suggestions will be valuable in improving the manuscript.

Best wishes

Michael Poulsen

Reviewer #2

(Remarks to the Author)

This manuscript is very well written and addresses a timely and important question about the link between personal and social immunity in eusocial insects. The data are presented in 3 parts using a comprehensive set of approaches. In the first part, the authors describe the immune machinery of this ant species using bioinformatic and genomic approaches. By comparing their data with those from *Drosophila*, they show that the *L. niger* genome encodes the complete NF- κ B pathway (the major systemic immune pathway in insects) and that it contains only 4 genes encoding antimicrobial peptides (AMPs), which are thought to be involved in systemic immunity. In the second part, they used functional approaches to show that the regulation of some of these genes depends on tissue damage and cues from two bacteria (G+ and G-), but not on cues from two entomopathogenic fungi. They then compared these responses with those of reproductive individuals and found some differences, particularly in the expression of AMP genes. In the third and final part, they investigated why the personal immune response is efficient against bacteria but not against fungi. They used experimental approaches and presented data showing that personal immunity could help them fight opportunistic infections from bacteria present on the cuticle, whereas social immunity could be efficient enough to limit the risk of exposure to fungi.

Overall, I found the study interesting, although the emphasis and strength of the different parts is uneven. The introduction is a bit unfocused on the main aim of the study (to test whether personal and social immunity are maintained because they target different types of pathogens). The first part seems robust (I cannot judge the accuracy of the method used as I am not familiar with genomic approaches) but is very descriptive. The second part provides a nice and robust way to test the function of the AMPs found in the first part (although the AMPs have not been synthesised and it is therefore still unclear whether they actually have an antimicrobial function and/or whether their regulation is actually linked to their antimicrobial function). The main problem with this part is the comparison with reproductive individuals, as it comes out of the blue and it is unclear what the reader can do with this information and how it relates to the main question. The last part is, in my opinion, the main selling point of the study. However, it is also the least convincing to me. The results of the proposed experiment are somewhat over-interpreted and the experimental design includes confounding factors that need to be ruled out before it can be robustly concluded that allogrooming is not induced by bacteria and that allogrooming is less efficient against bacteria than against fungi (see more details below).

In conclusion, I think that the study has the potential to make a nice and important contribution to the field, but the introduction needs to be more focused and the third part should include additional experiments to strengthen the conclusion.

More detailed comments can be found below:

L48: I am not sure that self-isolation can be defined as a collective defence, although it fits perfectly into the definition of social immunity.

L56-57: "... the expression of some immune genes has been reported to be induced..." (You need to mention that it is the expression that is induced, not the gene itself).

L61: To me, the introduction is very descriptive, a bit unfocused and, most importantly, lacks a clear prediction of why social and personal immunity would be expected to be involved against different types of pathogens. This is the main message of the study, but it is not introduced at all.

L63: It might be worth repeating that NF- κ B pathways are the main systemic immune pathways in insects.

L87: For clarity, would it be useful to say that it is for bacteria? It would contrast fungi and bacteria and make the last sentence of the following paragraph clearer for the reader.

L89-90: In the text, you only compare your data with *Drosophila*. So it is a bit of a stretch to say that the two pathways are similar to those of other hymenopterans. I think you need to detail the similarity to other Hymenoptera (and specify which Hymenoptera) to support this claim.

L235: I am being a bit picky here, but it does not mean that it is due to cuticle breaches. It could be due to the stress of

fighting, which interferes with the immune response and makes them more susceptible to the bacteria.

L259: I am a bit sceptical about the possibility of extending your conclusion that far. You tested one bacteria at one concentration (and two fungi at one concentration) and it is well-known that immunity often shows a dose-dependent response. To reach such a general conclusion, you would have needed to test (at least) several bacteria.

L267: Please add the value for the sham exposure.

L267: Unfortunately, there is also no difference in allogrooming between bacteria and fungi. Not sure if this is robust evidence for the effect of allogrooming against fungi and not bacteria.

L271: Self-grooming is another important mechanism by which an individual can remove external pathogens. Here you have no control-individuals maintained alone to determine whether the reported reduction is due to allogrooming rather than self grooming.

L275-282: Isolation not only prevents access to social immunity but also induces stress that can affect many physiological pathways, including immunity. The fact that workers survived a fungal infection less well when isolated than when in a group may also suggest that the stress of social isolation interferes with immune pathways that might be involved in fighting fungus (rather than bacteria). The experimental design is thus inaccurate to robustly support the proposed interpretation.

L291: You could use another word than "complex", or try to explain what you mean better with a longer sentence (or more words)?

L648: Is this the first time this approach has been used? If not, it would be nice to cite other papers that have used it (to reinforce its biological meaning).

Reviewer #3

(Remarks to the Author)

This is an interesting study on the immune response of the ant *Lasius niger* to bacterial and fungal pathogens. The paper is well written and the figures are clearly presented. Overall, I think this adds a lot to the field. I do have several questions.

1) The overall conclusions of the study are presented as erosion of individual immunity to fungi, while immunity to bacteria being maintained. However, all the data presented seems to show the immune response of *Lasius niger* to bacterial infections is exceedingly weak.

2) On lines 31-33, the authors state 'Overall, our results provide experimental confirmation of the functional erosion of the individual immune capacity against infections that are effectively handled by social immunity'. The results from Fig. 5F do not support social immunity being particularly effective against fungi.

3) Line 175, why didn't the authors use live spores of *Beauveria bassiana* in this study? I find it a major weakness that all this work involved a single fungal strain with lots of statements made across all fungi. This is particularly concerning given other studies (noted lines 342-345).

A few minor things.

Line 199 Change to 'independent'

Lines 205-206. This sounds a bit purposeful.

Lines 258-259. It is hard to see the case for systemic immunity against bacteria being 'effective' from this study.

Version 1:

Reviewer comments:

Reviewer #1

(Remarks to the Author)

The carefully revised manuscript addresses the concerns that I raised in my former review, and I also appreciate the authors' great efforts to address the requests from the other reviewer.

I only have very minor suggestions in the attached annotated version of the revised manuscript file.

Congratulations on an impressive piece of work that I think will greatly push the field of social immunity in insects forward.

Best wishes

Michael Poulsen

Reviewer #2

(Remarks to the Author)

The authors have properly addressed all the issues I raised with the first version of their study. The resulting manuscript is clearer and more efficient, and the take-home message is better supported by the data and analyses presented. I am happy to recommend publication of this paper, which is sure to attract a lot of attention from the community and be an important piece in our current understanding of the evolution of social immunity in insects.

Reviewer #3

(Remarks to the Author)

No remaining concerns or comments.

Response to reviewers

We are grateful to the Reviewers for their thorough reading of our work and for their insightful comments. We have made changes to the text and ran additional experiments as requested by the Reviewers, and we believe that the manuscript is greatly improved as a result. Please find below a point-by-point reply to each comment.

Reviewer #1 (expert in social immunity in ants):

The manuscript by Masson is a very welcome test of a so far poorly resolved hypothesis of immune gene erosion in social insects, addressing a fundamentally important aspect of the relative role of social and individual immunity in an ant species. The manuscript sets out in an extensive and convincing set of experiments to show the immune gene repertoire in *Lasius niger* and testing specifically the effectors against fungal and bacterial infections in biologically-relevant contexts. The authors show that individual immunity protects against opportunistic bacterial infections which are not covered by social immunity. By contrast, social immunity targets fungal infections, which do not elicit an effective individual immune response. This provides an important next step in understanding how millions of years of evolution of optimised immune targets have resulted in complementary actions of social and individual immunity.

The experimental work present in the paper is solid, all conclusions warranted, and I have no major reservations of the experimental work. However, I feel that the study could benefit from better comparisons across ant species to allow assessing how generalisable these findings are and how they relate to ant ecology. I suspect that complementarity might well be old in the group and with many genomes available, it is possible to, at least at the genome level, ascertain this based on immune pathway annotations. I recognise that performing comparative experimental work across ant lineages will likely not be possible, but the molecular signals based on genomics could provide the authors the ability to better assess (and discuss) how general these patterns are predicted to be and potentially identify ecologically-relevant differences across species (e.g., related to whether species are predatory, herbivores, or generalists as is alluded to in the discussion but never explicitly tested).

We are grateful for the positive evaluation of our work and the constructive review. We agree that further comparative analysis would provide a broader perspective on the paper. An extended ant comparative genomics analysis is currently being conducted as part of the GAGA project and is expected to be published soon in the consortium's flagship paper. This analysis will include immune pathways, and whilst the GAGA team shared their unpublished ant sequences with us, they did not wish us to duplicate their ongoing analysis.

Hence, to address the comment without overlapping with their work, we focused specifically on the GGBP family (microbe-associated molecular pattern receptors), as this is the main divergence point for *L. niger* compared to other Hymenoptera and conducted an orthology analysis across 163 ant genomes and other representative Hymenoptera. This new analysis confirmed that *L. niger* was missing a GGBP orthogroup and showed that this feature is shared by other *Lasius* species, including *Lasius flavus* and *Lasius neglectus*, but not by other Formicidae. This suggests that the genomic erosion of the GGBP family is not old, nor is it widely shared across ant lineages.

These new findings do not invalidate our hypothesis that individual immunity is redundant/plays a limited role against pathogens that are effectively targeted by social immunity, because genomic erosion is a far-downstream, non-obligatory consequence of functional redundancy and need not have occurred in all lineages. However, our new genomic analysis does not allow us to generalise our findings across ants and social insects in general. Furthermore, genomic erosion is too rare to allow

us to properly test for a relationship between ecology traits and the importance of individual immunity against fungal pathogens.

Furthermore, our new extended analysis indicated that ant GNBPs were not clearly resolved in relation to the *Drosophila* genes GNB1, 2, and 3. Therefore, we decided to moderate our interpretation and rephrase our conclusion to state that “the GNB family contains a reduced number of functional genes compared to other Hymenoptera” instead of explicitly naming GNB3.

The manuscript has hence been modified as follows:

- The first results subsection now includes the detailed orthology analysis results for GNBs (lines 87-94).
- The Methods section now includes a paragraph detailing the orthology analysis (lines 481-488).
- Supplementary Figure S1 now includes the complete phylogenetic tree of GNBs across Hymenoptera, showing the lack of *Lasius* GNBs in the second orthogroup.
- We have modified the text to avoid explicitly naming GNB3.
- We have clarified that our findings apply to black garden ants but not necessarily all ants by modifying the title and other relevant parts of the manuscript.
- The discussion part related to the lifestyle hypothesis (lines 361-374) has been rephrased to clarify that it is speculative at this point, based on our experimental data and literature rather than on a rigorous genomics analysis, and we call for further experimental comparative studies including more species to assess its validity. It now reads: “As *L. niger* mostly feeds on insect remains, nectar and honeydew, we assume that [...] ... Comparisons between *L. niger* and *M. analis* supports the idea that the likelihood of injuries may influence selection pressures for the evolution of bacteria-targeting social immune mechanisms, although further comparative studies will be required to verify the validity of this hypothesis”.

Similarly, while both *Pseudomonas* and *Metarhizium* are good model pathogens for experimental tests, and vast comparisons across pathogens obviously limited, the authors should at the least discuss how generalisable the conclusions are likely to be.

We agree that providing data on more model pathogens would greatly improve the biological significance of our conclusions. Therefore, we replicated the key results of Figure 5F and 5G (social protection against fungal but not bacterial infections) using a different set of microbes. We used the fungus *B. bassiana* and the bacterium Ecc15, which we used to characterise the immune response at the individual level in Figure 3C and Figure 2, respectively. This new experiment confirms that group living improves survival upon challenge with fungi but not bacteria. These results are now included in the manuscript at lines 270-274 and shown on a new Figure S5 and the methods section has been revised accordingly.

The revised manuscript now includes data obtained with four microbes (two bacteria and two fungi) for individual immunity (AMP gene induction, Figures 2 and S3) and for social immunity (Figures 4 and S5), which provides robust support for our main conclusion that individual immunity and social immunity are complementary and combat different classes of microbes.

Minor comments by line numbers:

Title: I would change “in ants” to “in an ant” or “in the common garden ant” to clarify that it is in one species and avoid overstating the findings.

We agree and have revised the title accordingly.

Line 58-59: I recognise the need for brevity but some detail would be nice for readers not already familiar with the literature.

We have modified the introduction as per Reviewer 2's recommendations, and consequently this statement no longer appears; hence we have not added detail.

Line 162. Missing 'of' before AMP induction **Done**.

Line 205: I think it should be 'and' instead of 'but' **Done**.

Line 225. "hereafter" instead of "thereafter"? **Done**.

Line 335: the authors come back to the point in the discussion that cuticle breaches may not be frequent in non-predatory ants, so a bit more caution here on the importance of breaches may be good to include. Has any work quantified the frequency and importance of breaches in nature?

The frequency of natural wounds remains an open question in the field of insect immunology, with no clear data available. To the best of our knowledge, no ongoing studies are being conducted on ants on this question.

Therefore, we rephrased the discussion as follows: 'As *L. niger* mostly feeds on insect remains, nectar, and honeydew, we assume that natural injuries are rare in this species' (L 361). This clarification aims to highlight that our qualitative estimate of the frequency of cuticle breaches in *L. niger* is an hypothesis we make based on the ant's lifestyle, rather than established scientific consensus.

Line 262: 'on' should be 'of' **Done**.

Line 271: what is known about cue differences between fungi and bacteria? I also do not see a discussion of (hypothesis for) why social and individual immunity may be subject to complementarity in ants.

Indeed, this point was not developed in the manuscript. Purified ergosterol, a lipid component of fungal membranes, is sufficient to elicit allogrooming in both ants (Stock et al. 2023 <https://doi.org/10.1038/s41559-023-01981-6>) and termites (Chen et al. 2022 <https://doi.org/10.1111/1744-7917.13055>), suggesting that it is a fungus-specific chemical cue broadly recognised by social species. However, there are likely other cues involved that remain unidentified. Regarding bacteria, no specific cue has been identified thus far due to the limited number of studies available.

Our data does not allow us to draw any conclusions about the cues involved in triggering allogrooming for either type of pathogen. Therefore, we modified this part of the results to remove the concept of "cues" from the results and now state that "[...] allogrooming is not increased by bacterial exposure, [...] allogrooming levels elicited by fungi" (L. 260-262) to better reflect our data. The concept of complementarity between social and individual immunity is based on our observation that each set of mechanisms confers protection preferentially against pathogens that the other set does not (individual immunity protects against bacteria but not fungi, while social immunity protects against fungi but not bacteria). This point is developed in the discussion at lines 330-384.

Line 275: I would cut 'Finally,' **Done**.

Line 301: cut 'with'? **Done**.

Line 149 and elsewhere: giving the comparisons in parentheses is perhaps not needed as it's imperative that they represent a comparison of an experimental vs a sham treatment. Instead, I recommend giving the clear test in the text and then stating the type of test in parentheses. Thank you, you are right that this level of detail was unnecessary for some of the comparisons described. We have now deleted explicit mentions of the contrast tested wherever it was obvious which comparison the statistical analysis in parentheses referred to (e.g. lines 149, 152, 153).

Line 352: why were these components not evaluated as well, what would it take, and what would the authors predict? While leaving these out does not take away from the clear patterns that the authors present, it prevents a more complete picture of the relative roles of (clearly important) immune mechanisms in ants.

That is a legitimate concern, as insect immunity is indeed not limited to systemic AMP production. Unfortunately, we were not able to investigate melanization, cellular immunity, and other immune mechanisms in-depth because of time constraints, as the lead author has now left academia. Instead, we used dedicated methods to minimize the involvement of these other mechanisms in our experiments (samples were centrifuged to remove hemocytes and treated with PFU to block melanization, as indicated in the Methods on line 597-601), which should ensure that our conclusions about NF- κ B pathways are valid. Nonetheless, we have added a cautionary statement in the discussion on lines 346-347: 'Our results, however, do not exclude the possible involvement of other individual immunity mechanisms active against fungi, such as melanization or cellular immunity'.

The final paragraph of the discussion starting at Line 381 should be toned down to better represent the findings, avoid overstatements and present limitations. For example, stating that the work "...illustrates the diversity of immune defense mechanisms across this group, in the light of their ecology and evolutionary history limitations" based on a single ant species and two pathogens is a gross overstatement. It would be much more useful for readers if the authors focused on what this tells us, where the limitations are, and what incisive new questions with predictions are (e.g., patterns across the ant phylogeny).

We agree with this recommendation and have removed the overstatement highlighted by the Reviewer. We have also toned down other parts of the discussion related to species ecology as detailed above. Furthermore, the closing paragraph (L. 375-384) now emphasizes the unique evolution of social insects' immunity and the need for dedicated functional research to replace assumptions drawn from non-social insect models and to better understand the diversity of immune pathways across social insects.

Across all figure legends, specific methods used (e.g., qPCR) should be mentioned as it is not clear from the figures themselves and sample sizes should be provided.

This information has been added across all figure legends.

Figure S4 is not cited in the text as far as I can tell and it doesn't seem to be important for the paper.

This figure is cited in the Methods section (now Figure S6, L. 415). While we agree that it may not be crucial for the majority of readers, it provides concrete evidence about the sequencing quality, and we believe it could be of interest to specialists in genome assembly methods.

I hope these suggestions will be valuable in improving the manuscript.

They definitely were, and we are grateful for such a thorough reading and constructive review!

Best wishes
Michael Poulsen

Reviewer #2 (expert in social immunity in insects):

This manuscript is very well written and addresses a timely and important question about the link between personal and social immunity in eusocial insects. The data are presented in 3 parts using a comprehensive set of approaches. In the first part, the authors describe the immune machinery of this ant species using bioinformatic and genomic approaches. By comparing their data with those from *Drosophila*, they show that the *L. niger* genome encodes the complete NF- κ B pathway (the major systemic immune pathway in insects) and that it contains only 4 genes encoding antimicrobial peptides (AMPs), which are thought to be involved in systemic immunity. In the second part, they used functional approaches to show that the regulation of some of these genes depends on tissue damage and cues from two bacteria (G+ and G-), but not on cues from two entomopathogenic fungi. They then compared these responses with those of reproductive individuals and found some differences, particularly in the expression of AMP genes. In the third and final part, they investigated why the personal immune response is efficient against bacteria but not against fungi. They used experimental approaches and presented data showing that personal immunity could help them fight opportunistic infections from bacteria present on the cuticle, whereas social immunity could be efficient enough to limit the risk of exposure to fungi.

Overall, I found the study interesting, although the emphasis and strength of the different parts is uneven. The introduction is a bit unfocused on the main aim of the study (to test whether personal and social immunity are maintained because they target different types of pathogens). The first part seems robust (I cannot judge the accuracy of the method used as I am not familiar with genomic approaches) but is very descriptive. The second part provides a nice and robust way to test the function of the AMPs found in the first part (although the AMPs have not been synthesised and it is therefore still unclear whether they actually have an antimicrobial function and/or whether their regulation is actually linked to their antimicrobial function).

We sincerely thank Reviewer 2 for their appreciation of our work and their constructive criticism.

The main problem with this part is the comparison with reproductive individuals, as it comes out of the blue and it is unclear what the reader can do with this information and how it relates to the main question.

We agree that this was a weakness of the original manuscript, and we had actually hesitated whether or not to include it in the submitted version. To be candid, these results were obtained as part of exploratory research, and were included in this manuscript because the first author has left academic research and would not have been able to publish them in a separate standalone article. The question of the immune response of reproductive individuals in social insects is a timely topic, and we believe that these results will be of interest to many groups working on this subject; however, we agree that they were not elegantly integrated into the narrative in the original manuscript, and not required to support our main conclusions. Therefore, we have significantly shortened the results section related to reproductives and merged it with the section about the workers' individual response (see lines 183-193), and we have moved the corresponding figure to the supplementary material (now Figure S4). We believe that the flow of the manuscript now works much better, and readers interested in this point will still be able to access the data in the SI, so we hope this is an acceptable compromise.

The last part is, in my opinion, the main selling point of the study. However, it is also the least convincing to me. The results of the proposed experiment are somewhat over-interpreted and the experimental design includes confounding factors that need to be ruled out before it can be robustly concluded that allogrooming is not induced by bacteria and that allogrooming is less efficient against

bacteria than against fungi (see more details below).

We have acknowledged this comment and conducted additional experiments and analyses to strengthen our conclusions, detailed in the point-by-point replies below.

In conclusion, I think that the study has the potential to make a nice and important contribution to the field, but the introduction needs to be more focused and the third part should include additional experiments to strengthen the conclusion.

More detailed comments can be found below:

L48: I am not sure that self-isolation can be defined as a collective defence, although it fits perfectly into the definition of social immunity.

Indeed, we agree that listing 'self'-isolation as part of 'collective' defenses sounded counter-intuitive. We have thus replaced 'collective' defenses with 'social' defenses. Furthermore, since isolation can be either voluntary or enforced by nestmates, we have also removed 'self-' and opted for the broader term 'isolation of contaminated workers'.

L56-57: "... the expression of some immune genes has been reported to be induced..." (You need to mention that it is the expression that is induced, not the gene itself).

Thank you; this has now been corrected.

L61: To me, the introduction is very descriptive, a bit unfocused and, most importantly, lacks a clear prediction of why social and personal immunity would be expected to be involved against different types of pathogens. This is the main message of the study, but it is not introduced at all.

Thank you for pointing this out; we have reworked the introduction accordingly. We now clearly formulate our hypothesis on line 56: « Here we hypothesize that individual immunity in social insect species have evolved towards a reduced, but still functional arsenal specialized in fighting microbes that are not targeted by social immunity», and we modified the last paragraph of the introduction to articulate it around this hypothesis (Lines 59-66).

We also removed the part describing examples of why research on the matter has been hampered thus far (former lines 58-61), as it was not required for the readers to understand the background or the hypothesis.

We hope that these modifications now make the introduction more structured as follows:

[1] Premises: social immunity is predicted to release selection on individual immunity genes, yet core individual immunity genes are conserved in social species.

[2] Hypothesis: individual immunity in social species is not diminished but rather specialized, and thereby it complements social immunity.

[3] Summary of the methods, results, and conclusions with regards to the hypothesis.

L63: It might be worth repeating that NF_κB pathways are the main systemic immune pathways in insects.

This is now repeated on L. 60-61.

L87: For clarity, would it be useful to say that it is for bacteria? It would contrast fungi and bacteria and make the last sentence of the following paragraph clearer for the reader.

Thank you for this suggestion, we agree that this will help with clarity of the argument. Accordingly, we have now rewritten the sentence to specify which microbe each molecular pattern belongs to. The sentence now reads: 'In *Drosophila*, the extracellular part of the pathway is immunity-specific and recognizes microbes through the pattern-recognition receptors GNB3 for fungi β -glucans and GNB1 and PGRP-SA for bacterial (lys)-type PGN' (L. 85 in the new version).

L89-90: In the text, you only compare your data with *Drosophila*. So it is a bit of a stretch to say that the two pathways are similar to those of other hymenopterans. I think you need to detail the similarity to other Hymenoptera (and specify which Hymenoptera) to support this claim.

This is a very good point, the original manuscript indeed did not cite the relevant studies that support the similarity with Hymenopterans. To address this, we have now added the following sentence (L. 73-74): "*Although variability exists, the core component of these pathways and largely conserved across insects, including Hymenopterans such as bumble bees, honeybees and ants*^{10,20,46}". This new statement should clarify that pathway similarity concerns core genes (i.e., excluding regulators, which are more prone to variability across species), and includes references that broadly cover the Hymenoptera order. We also referenced these sources again in the concluding paragraph of this section (L. 95): '[...] similar to that of other Hymenopterans^{10,20,46}'.

L235: I am being a bit picky here, but it does not mean that it is due to cuticle breaches. It could be due to the stress of fighting, which interferes with the immune response and makes them more susceptible to the bacteria.

It is true that social stress could lead to a similar phenotype, as shown in *Harpegnathos saltator* by Schneider et al. (2016); thank you for pointing this out. In our case, as the decreased survival phenotype can be recapitulated by a sterile wound without a fight (antenna cutting), we believe that fight-related cuticle lesions are the most probable cause, but we agree that our data do not allow us to formally rule out any effect of social stress. We have thus modified this paragraph to acknowledge this alternative explanation and cite the work by Schneider et al. (2016). The text now reads as follows:

'[...] survival to an external bacterial challenge appears to be jeopardized by fighting. Although we cannot rule out that this may be due to combat-induced stress³⁹, it is more likely a consequence of fight-related cuticle lesions, as these results closely mirror those produced by experimenter-inflicted sterile injury (Figure 4C).'

L259: I am a bit sceptical about the possibility of extending your conclusion that far. You tested one bacteria at one concentration (and two fungi at one concentration) and it is well-known that immunity often shows a dose-dependent response. To reach such a general conclusion, you would have needed to test (at least) several bacteria.

This point has been raised by Reviewer 1 as well, and we agree that testing additional microbes would add significant value to our data.

We were unfortunately unable to repeat the hemolymph trials with additional bacteria and/or more concentrations because the lead author has left academia and the specialized training required to carry out these trials is no longer available in our lab. Furthermore, these trials would require a substantial number of queens collected in summer then left to incubate claustrally for several months, which we do not have at our disposal.

However, we did expand the survival experiment to include one additional bacterium and one additional fungus. We now present survival data showing that social immunity does not confer significant protection against the bacterium Ecc15 but does confer protection against the fungus *B. bassiana*. These new results are shown in a new Figure S5 and described at lines 270-275.

Now, both our key experiments (AMP induction and individual vs social survival) provide data using two bacteria and two fungi, indicating that our conclusions can be generalised to some extent.

L267: Please add the value for the sham exposure.

We have added this information on L. 254 (10.4 events/focal ant).

L267: Unfortunately, there is also no difference in allogrooming between bacteria and fungi. Not sure if this is robust evidence for the effect of allogrooming against fungi and not bacteria.

We sincerely appreciate this and the next comment, which led us to thoroughly revise our analysis.

Initially, we had annotated each allogrooming event without considering the number of nestmates involved. This approach did not account for multiple allogrooming events, where several nestmates engage in allogrooming the same focal ant at the same time, or where a continuous string of ants engaged in successive but overlapping grooming sessions; so, for example if five nestmates engaged in grooming the focal ant simultaneously, this was counted as a single allogrooming event; or if ten ants repeatedly started and stopped grooming the focal ant while another ant was engaged in continuous grooming, this was also counted as a single allogrooming event.

In the revised manuscript, we conducted a new analysis on the same dataset, now accounting for multiple allogrooming events (in the example given above, five nestmates engaged in grooming the focal ant simultaneously were counted as five grooming events). We believe this approach is more realistic as it better reflects the frequency and intensity of allogrooming.

We then repeated the statistical analysis as previously done. The general trends remained the same; but the difference between fungi-exposed ants and bacteria-exposed ants is now significant.

This new analysis replaces the former one in the manuscript. Figure 4C has been updated accordingly, and the methods section has been revised to reflect these changes.

All original videos will be uploaded on a public repository to support our new analytical approach.

L271: Self-grooming is another important mechanism by which an individual can remove external pathogens. Here you have no control-individuals maintained alone to determine whether the reported reduction is due to allogrooming rather than self grooming.

This is a very valid concern, and it is indeed likely that self-grooming was at least in part responsible for pathogen removal in our experiment; thank you for pointing this out.

Controlling for self-grooming experimentally is actually not trivial because isolated ants are known to greatly increase the frequency and duration of self-grooming behaviours compared to ants kept in groups. Therefore, direct comparisons with isolated control-ants would not allow us to determine the relative importance of self vs allogrooming in ants kept in groups, which is why our experimental design did not include such control-individuals.

However, the Reviewer is right that we do need to quantify the relative importance of self-grooming in our experimental conditions and show that self-grooming is not a likely cause for the differential microbe removal efficiency observed between fungi and bacteria. Fortunately, as we still had the source videos, we were able to quantify self-grooming in our dataset using the same behavioral annotation approach as for allogrooming. We found three lines of evidence supporting that self-grooming is not responsible for the trends observed in our data:

1. The number of self-grooming events was less than 18% of all grooming events in both pathogen treatments, indicating that the behavioral response of the group to pathogen exposure of one ant mainly consists of allogrooming elicited in nestmates.
2. The number of self-grooming events did not differ between treatments (LMM $\chi^2 = 4.55$, $df = 2$, $p = 0.103$) and, therefore, is unlikely to explain the significant difference in microbial load reduction.
3. There was a high correlation between load reduction (qPCR data from Figure 4D and E) and the number of allogrooming events (Figure 4C; Pearson's product-moment correlation, both pathogen treatments pooled: $t = 3.352$, $df = 29$, $p\text{-value} = 0.002$) but no correlation between load reduction and the number of self-grooming events ($t = -0.41343$, $df = 29$, $p\text{-value} = 0.682$).

As an illustration, please see below the allogrooming and self-grooming counts plotted side-by-side. Collectively, these lines of evidence support the idea that self-grooming is not significantly involved in the differential decrease in microbial load in our experiment.

L275-282: Isolation not only prevents access to social immunity but also induces stress that can affect many physiological pathways, including immunity. The fact that workers survived a fungal infection less well when isolated than when in a group may also suggest that the stress of social isolation interferes with immune pathways that might be involved in fighting fungus (rather than bacteria). The experimental design is thus inaccurate to robustly support the proposed interpretation.

This is a valid concern, as it has been shown that isolation stress interferes with immune gene expression in mammals. However, such observations have not been replicated in insects. To the best of our knowledge, only a single study by Scharf et al. (2021) found a link between social isolation and immune gene expression in ants. However, the genes identified in that study were automatically labelled as “immune” genes based on GO annotations, but they were actually primarily related to metabolism or general stress, and included only distal and/or indirect regulators of the immune response (Adenosine deaminase 2, Aromatic-L-amino-acid decarboxylase, cytochromes, heat-shock proteins, Charybde, etc.). No immunity-specific gene was shown to be affected in that study. A more recent study by Koto et al. (2023) also failed to detect any transcriptional evidence of an effect of isolation on the expression profile of immune genes.

It should also be pointed out that whilst the cues which trigger the two main immune pathways in insects (Toll and Imd) are pathogen-specific, the signalling and effector parts of these pathways are shared across pathogens and would thus be similarly affected by social isolation. While we cannot

fully exclude the possibility that isolation stress interferes with the detection of fungal-specific cues more than it interferes with the detection of bacterial-specific cues, we find this unlikely as our genomic analysis suggests that the detection of fungal cues is likely to be impeded in *Lasius* ants by the lack of a GGBP orthogroup regardless of social context.

Finally, previous studies have demonstrated that the benefit of group living on survival to fungal challenge is mainly due to physical spore removal via allogrooming and chemical disinfection with formic acid, independently of social isolation (Tragust et al 2013).

In conclusion, the current state of knowledge in the field leads us to conclude that the lower resistance to fungal infections in isolated ants is most likely due to a lack of social protection, as discussed in our manuscript.

L291: You could use another word than "complex", or try to explain what you mean better with a longer sentence (or more words)?

We agree that this wording was too vague and we clarified by replacing "less complex" by "depauperate", which is the term used in the cited literature to describe the evolutionary state of honeybee immune pathways (Barribeau et al. 2025; <https://doi.org/10.1186/s13059-015-0628-y>).

L648: Is this the first time this approach has been used? If not, it would be nice to cite other papers that have used it (to reinforce its biological meaning).

We chose this approach because we believed that converting a non-visual unit (mass of DNA) into a visual one (number of microbe cells) would be more meaningful to a broader audience.

Similar approaches have been widely used in the literature, especially in plant microbiology (see for example

Badri et al. 2016 <https://link.springer.com/article/10.1007/s00572-016-0708-1>;

Helin et al. 2017 <https://doi.org/10.5194/acp-17-13089-2017>;

Brandendurg et al. 2024 <https://doi.org/10.3390/jof10040237>;

Clark et al. 2024 <https://apsjournals.apsnet.org/doi/10.1094/PHYTOFR-04-23-0045-FI>)

and in the field of insect social immunity (Stroeymeyt et al. 2018 <https://doi.org/10.1126/science.aat4793>).

The conversion details are different across studies, but always based on simple proportionality (in our case, using published genome sizes for *M. brunneum* and *P. entomophila*). See e.g. below a comparison between our approach and the approach used in Stroeymeyt et al. 2018; the two approaches produce results within the same of order of magnitude (differing by a factor of 1.5).

Our conversion method is detailed in the Methods section (L. 651-654) and the raw data is available in a public repository, allowing readers to easily access the raw values expressed in nanograms of DNA for comparative studies.

Cell quantification using the conversion approach used in the manuscript (left) and the conversion factor used in Stroeymeyt et al. 2018 (right).

Reviewer #3 (expert in ant-microbe symbiosis and molecular ecology):

This is an interesting study on the immune response of the ant *Lasius niger* to bacterial and fungal pathogens. The paper is well written and the figures are clearly presented. Overall, I think this adds a lot to the field. I do have several questions.

We are very grateful for this positive evaluation of our work and address these questions below:

1) The overall conclusions of the study are presented as erosion of individual immunity to fungi, while immunity to bacteria being maintained. However, all the data presented seems to show the immune response of *Lasius niger* to bacterial infections is exceedingly weak.

It is true that the levels of AMP induction (hym: x5000 in response to Ecc15 relative to unchallenged; x3000 in response to *M. luteus* relative to unchallenged) are rather weak. However, these levels were clearly detectable and statistically significant compared to sham-injections, and our in vitro assays of the antimicrobial activity of challenged hemolymph extracts confirmed that this response is functional and successful at reducing the viability of a bacterial species (Fig 4A), but not a fungal species (Fig 4B).

It would be of great interest to quantify the exact level of protection it confers and compare it with that of other models. However, this would require genetic tools to impair the signaling pathways, which is currently out of reach in non-clonal ants, unfortunately (Sieriebriennikov et al. 2021 <https://doi.org/10.1016/j.tig.2021.05.005> ; Konu et al. 2023 <https://doi.org/10.1111/imb.12809>).

2) On lines 31-33, the authors state 'Overall, our results provide experimental confirmation of the functional erosion of the individual immune capacity against infections that are effectively handled by social immunity'. The results from Fig. 5F do not support social immunity being particularly effective against fungi.

That is a good point. First, we would like to mention that the intensity of the effect of allogrooming on survival is known to vary greatly across host species and pathogen strains (see e.g. Pontes Stefanelli et al. 2021; <https://doi.org/10.3390/insects12010010>), as well as laboratories, experimental protocols (e.g. group size or pathogen dose) and past history of colonies (e.g. colonies raised in the lab or ants collected in the field). The effect sizes measured in our study are indeed at the low end of effect sizes reported in the literature, but they are within the observed range, and the direction of the effect (protection against fungi) aligns with multiple studies conducted with various fungal strains (see Cremer et al. 2019 for a review and extended list of references).

Despite the variation in reported effect sizes of allogrooming, the effectiveness of social immunity against fungi is very well established in the field (Cremer et al. 2017; <https://doi.org/10.1146/annurev-ento-020117-043110>; Cremer 2019; <https://doi.org/10.1016/j.cub.2019.03.035>), and it arises from a broad panel of mechanisms; social care via allogrooming (which is the main mechanism occurring in small-group survival experiments such as the one we carried out) is only one of these mechanisms. As small groups are known to display weaker social immunity phenotypes than larger groups, in natural conditions we would expect even a weak effect of social care to be amplified in larger groups; and this effect would be further completed by additional social immunity mechanisms which we did not measure here.

However, we accept that our results do not directly illustrate the high effectiveness of social immunity against fungi and have therefore removed the word "effectively" from that sentence.

3) Line 175, why didn't the authors use live spores of *Beauveria bassiana* in this study? I find it a major weakness that all this work involved a single fungal strain with lots of statements made across all fungi. This is particularly concerning given other studies (noted lines 342-345).

The issue with using live *Beauveria bassiana* spores is that they were almost immediately lethal when administered via direct injection, even at low doses. This prevented us from measuring AMP gene induction by injection of live *B. bassiana* spores 24 hours after injection, as we did for all other pathogens. For this experiment, we preferred to use the same protocol as for the other pathogens and so stick with injections (rather than resorting to external exposure) to ensure comparability of the molecular response between pathogens without variability arising from the cuticle penetration process by the fungus. Importantly, however, it should be noted that the main fungal molecular patterns that activate the Toll pathway, β -glucans, are heat-stable, and that heat-killed fungi are known to elicit this pathway in a similar way as live fungi. As heat-killed fungi lead to significantly higher survival rates, they allow molecular analyses of the immune response some time after injection, which is impossible with live spores of highly virulent strains.

Heat-killed spores have been used in previous studies for these exact reasons, including in seminal papers deciphering *Drosophila* immune pathways, so this is a well-established, well-accepted method in the field of insect immunity. For example, Gottar et al. 2006

<https://doi.org/10.1016/j.cell.2006.10.046> proved the involvement of GNB3 in fungal recognition using heat-killed *Candida albicans*. We have added this reference to the Methods section together with a short statement explaining that this is a valid method.

On a side note, it is important to point out that most articles on social immunity in ants have used *Metarhizium brunneum* instead of *Beauveria bassiana*, and that we were able to use live *M. brunneum* spores in this experiment, which allows for easy comparison of our results with those in the literature.

A few minor things.

Line 199 Change to 'independent'.

This has been corrected.

Lines 205-206. This sounds a bit purposeful.

We agree and we have changed the wording to the following:

'Rampant bacterial infections in wild colonies are rarely observed³⁹, yet the cuticle of several ant species is known to be frequently contaminated by pathogenic bacteria^{40,41}.'

Lines 258-259. It is hard to see the case for systemic immunity against bacteria being 'effective' from this study.

We agree and have changed the text to: 'This confirms that triggering systemic immunity confers antibacterial, but not antifungal, activity to the hemolymph,' to better reflect the data.